# Strategic Multi-Armed Bandit Problems Under Debt-Free Reporting

**Ahmed Ben Yahmed**
CREST, ENSAE, Palaiseau, France
Criteo AI Lab, Paris, France
FairPlay joint team
a.benyahmed@criteo.com

**Clément Calauzènes**
Criteo AI Lab, Paris, France
FairPlay joint team
c.calauzenes@criteo.com

**Vianney Perchet**
CREST, ENSAE, Palaiseau, France
Criteo AI Lab, Paris, France
FairPlay joint team
vianney.perchet@normalesup.org

## Abstract

We consider the classical multi-armed bandit problem, but with strategic arms. In this context, each arm is characterized by a bounded support reward distribution and strategically aims to maximize its own utility by potentially retaining a portion of its reward, and disclosing only a fraction of it to the learning agent. This scenario unfolds as a game over $T$ rounds, leading to a competition of objectives between the learning agent, aiming to minimize their regret, and the arms, motivated by the desire to maximize their individual utilities. To address these dynamics, we introduce a new mechanism that establishes an equilibrium wherein each arm behaves truthfully and discloses as much of its rewards as possible. With this mechanism, the agent can attain the second-highest average (true) reward among arms, with a cumulative regret bounded by $\mathcal{O}(\log(T)/\Delta)$ (problem-dependent) or $\mathcal{O}(\sqrt{T \log(T)})$ (worst-case).

## 1 Introduction

The multi-armed bandit (MAB) problem serves as a fundamental modeling framework for exploring the interplay between a decision-making agent (or player) and a set of arms. Its primary aim is to enable the player to learn the most favorable sequence of decisions over time. In formal terms, we consider a scenario involving $K$ arms, where, at each time step $t$, the player selects one arm $k_t$ from the set $\{1, 2, \ldots, K\}$ and receives a reward, denoted as $r_{k_t, t}$. These rewards can either be generated stochastically or determined adversarially. The player's overarching objective is to strike a balance between two key aspects: exploration and exploitation. Initially, he must explore all arms adequately to gain confidence in identifying the best-performing arm. Once this is accomplished, he focuses on exploiting this knowledge to maximize his cumulative reward over subsequent rounds. Established algorithms for this problem ensure that the player's performance is nearly as good as selecting the best individual arm, thus minimizing his regret [1, 2, 3]. This modeling framework has broad real-world applications, including domains such as clinical trials [4, 5], recommender systems [6], and resource allocation problems [7]. However, in many of these scenarios, arms are traditionally viewed as passive entities, faithfully reporting their generated rewards $r_{k_t, t}$ to the player.

38th Conference on Neural Information Processing Systems (NeurIPS 2024).

This perspective leaves out an array of dynamic agency dilemmas, where the player selects one of the $K$ agents (arms) at each time step to execute a task on his behalf, with the associated cost remaining hidden from the player due to his limited domain or market knowledge. Essentially, the player is uncertain about the precise costs or returns until the task is completed, and the agent has substantial freedom to set these ex-post [8]. In this context, it is reasonable to assume that arms are strategic and may act in their self-interest. Conceptually, arms can report a value, denoted as $x_{k_t,t}$, which may differ from the actual reward $r_{k_t,t}$, retaining in the process a net utility of $r_{k_t,t} - x_{k_t,t}$. This strategic scenario introduces a game-like dynamic, creating a competition of objectives between the player, aiming to minimize his regret and the arms, wishing to maximize their own utilities.

Motivated by numerous real-world applications where, by design, strategic arms operate under restricted payment conditions, our study focuses on debt-free reporting. In this setting, arms cannot report values higher than the observed reward. An illustrative example pertains to scenarios with binary variables, where declaring a fake failure is possible but creating a fake success is impossible. This situation is evident in e-commerce, where advertisers receive rewards for successful ad campaigns that result in a sale after a click. Notably, an e-commerce platform may choose to conceal a sale, but it cannot fabricate one. This setting is also referred to as budget-balance in the repeated trade literature [9, 10], requiring $x_{k_t,t} \leq r_{k_t,t}$ for all $t$, or equivalently, $(x_{k_t,t}, r_{k_t,t})$ belonging to the upper triangle $\mathcal{T} = \{(a,b) \in \mathcal{S}^2 \mid a \leq b\}$, where $\mathcal{S}$ represents the variables' space.

## 2 Problem Statement

The concept of the strategic multi-armed bandit builds upon the foundation of the classic multi-armed bandit problem, introducing a novel element wherein the arms possess the capability to retain a portion of the reward for themselves. Within this framework, we contemplate a collection of $K$ stochastic arms denoted by $k \in \{1, \ldots, K\}$, where each arm possesses a bounded reward distribution $D_k$ supported on $[0, 1]$. Each arm is distinguished by its reward mean, denoted as $\mu_k$. To maintain generality without loss of clarity, we assume, for the presentation and the analysis, an ordering such that $\mu_1 \geq \mu_2 \geq \cdots \geq \mu_K$ (unknown to the player). In each round $t$, the *player* selects an arm denoted as $k_t$. This arm $k_t$ then observes a reward $r_{k_t,t} \in [0,1]$. Subsequently, the *arm* reports a quantity $x_{k_t,t}$ to the player and retains $r_{k_t,t} - x_{k_t,t}$ as its own utility. In this scenario, it is important to note that solely arm $k_t$ possesses knowledge of the actually observed reward $r_{k_t,t}$, whereas the player is privy only to $x_{k_t,t}$ and remains unaware of the withheld portion. Hence the decision of the player is based on the information gathered until time $t$ which can formally encoded in $\mathcal{F}_{P,t} = \{k_s, x_{k_s,s}\}_{s<t}$ implying that $k_t$ is $\sigma(\mathcal{F}_{P,t})$ measurable. Conversely, the arm $k_t$'s available information is succinctly represented by $\mathcal{F}_{k_t,t} = \{k_s, r_{k_s,s}, x_{k_s,s}, k_t, r_{k_t,t}\}_{s<t:k_s=k_t} \cup \{k_s\}_{s<t}$ and $x_{k_t,t}$ is $\sigma(\mathcal{F}_{k_t,t})$ measurable. Specifically, this indicates that the tacit model is being utilized, wherein the arms are informed about the pulled arms at each round and only observe the reward upon being chosen.

Furthermore, we operate under the debt-free reporting assumption: arms are prohibited from incurring negative balances when reporting $x_{k_t,t}$, implying the following constraint $0 \leq x_{k_t,t} \leq r_{k_t,t} \ \forall t \geq 1$. In the context of strategic bandits, the player has the option to allocate a bonus $\Psi_k$ to each arm $k \in \{1, \ldots, K\}$ at the end of the game. This bonus serves as an incentive to encourage arms to provide truthful information, for instance. It can take various forms, including additional rounds of pulling as in [8] or simply a bonus payment awarded to the arms, similar to payments in the usual mechanism design for procurement auctions [11]. In this work, the latter option has been chosen for use, i.e. a bonus payment-based algorithm. Therefore, the interaction between the player and the arms can be encapsulated in the Model 1.

---

**Model 1:** The Strategic Multi-Armed Bandit Problem

---

1   Player commits to an algorithm $\mathcal{A}$, which is public to the arms
2   **for** $t = 1, \ldots, T$ **do**
3      Player selects arms $k_t \in \{1, \ldots, K\}$ according to the chosen algorithm $\mathcal{A}$
4      Arm $k_t$ observes reward $r_{k_t,t} \sim D_{k_t}$
5      Arm $k_t$ reports $x_{k_t,t} \in [0, r_{k_t,t}]$ to the player
6   **end**
7   Player assigns bonuses $\Psi_k$ to each arm $k \in \{1, \ldots, K\}$

---

## 2.1 Arm Utility and Subgame Perfect Equilibrium Among Arms

Let us denote by $\mathcal{P}$, the set of all possible arm strategies, by $\boldsymbol{\pi}_k = (\pi_{k,t})_{t:k_t=k} \in \mathcal{P}$ the strategy of arm $k$ and by $\boldsymbol{\pi} = (\boldsymbol{\pi}_1, \ldots, \boldsymbol{\pi}_k)$ the strategy profile of the arms. Explicitly, $\mathcal{P}$ is the set of mappings from the past history $\bigcup_{s=1}^{T} \left\{ [0,1]^{2s} \times \{0,1\}^T \right\}$ to $[0,1]$. Here, $[0,1]^{2s}$ denotes the accumulated observed and announced rewards, and $\{0,1\}^T$ indicates the rounds during which this arm has been selected thus far. Let $x_k \in \mathbb{R}_\star^T$ (and $r_k \in \mathbb{R}_\star^T$) be the sequence of the reported values (and rewards, respectively) by arm $k$ across $T$ rounds where $\mathbb{R}_\star = \mathbb{R} \cup \{\star\}$, and the symbol '$\star$' is used for rounds when the arm is not observed. We also refer to $x_k$ as the path of arm $k$ under strategy $\boldsymbol{\pi}$. Let $x_{-k}$ denote the $K-1$ paths of all arms except $k$. As a result, the utility associated with arm $k$ is defined in an ex-post fashion as follows:

$$\mathcal{U}_k(\boldsymbol{\pi}_k, \boldsymbol{\pi}_{-k}) = \sum_{t=1}^{T} (r_{k_t,t} - x_{k_t,t}) \cdot \mathbb{1}_{[k_t=k]} + \Psi_k(x_k, x_{-k}) \quad {}^{(1)} \tag{1}$$

which is the cumulative savings over rounds plus the corresponding assigned bonus (we slightly abused notations here, as $\Psi_k$ only depends on the observations of the player, and not the full vector $x_k$). Given that each arm aims to maximize its own utility, we introduce the solution concept of subgame perfect Nash Equilibrium (SPE) among arms. A strategy profile $\boldsymbol{\pi}^\star$ is a SPE if $\pi_k^\star$ is an optimal strategy for any arm $k$, considering any history and given any strategy $\boldsymbol{\pi}_{-k} \in \mathcal{P}^{K-1}$ adopted by other arms, at any time $t$.

## 2.2 Player's Strategic Regret

In the strategic scenario, the player, under the most unfavorable circumstances, cannot guarantee to achieve gains surpassing $\mu_2$. This limitation arises from the fact that the optimal arm merely requires a marginally higher reported value than the second-best arm to be chosen as the superior option by the player (see Section 2.3 for more details). In addition, the bonus allocated to the arms is deducted from the player's total revenue; hence, these bonuses are factored into the regret definition. Therefore, the regret, arising from the cumulative expected discrepancy between $\mu_2$ and the value reported by the selected arm $k_t$ over $T$ rounds, along with the bonus values, is defined as follows:

$$\mathcal{R}_T = \mathbb{E}\left[ \sum_{t=1}^{T} (\mu_2 - x_{k_t,t}) + \sum_{k=1}^{K} \Psi_k(x_k, x_{-k}) \right] \tag{2}$$

## 2.3 Related Work

A simplified instance of the strategic bandit problem has been examined within the context of the principal-agent problem in contract theory [12, 13]. In this analogy, a player engages an arm with greater expertise in a specific domain to perform work on his behalf. However, the arm may exploit the player's lack of knowledge to maximize its own utility and accumulate private savings. Consequently, the strategic bandit problem can be viewed as a principal-multiagents problem. Additionally, drawing inspiration from the field of online advertising, several studies have delved into dynamic mechanisms to address similar scenarios [14, 15, 16, 17, 18, 19, 20]. For instance, [21] examined auctions with reserve prices involving strategic myopic buyer.

In particular, [22] studied strategic arms but with a distinct utility function that depends solely on the number of pulls. In this scenario, each strategic arm aims to maximize the total number of rounds it is selected rather than cumulative savings. The authors demonstrate that stochastic MAB algorithms are robust to strategic manipulation, allowing the player to achieve $\mu_1 T$ with a sublinear regret. Under their utility definition, an arm is content to report the entirety of its reward as long as it is selected, justifying their use of the best mean $\mu_1$ as a comparator for regret.

A comparable study was conducted by [23], albeit with a variation in the arm's utility function, which incorporates the number of pulls alongside savings. The primary difference lies in their incorporation of a non-empty set of truthful arms, among other considerations. In this context, where arms are informed about competition and with additional assumptions conducive to specific MAB algorithms, they establish the resilience of these algorithms to strategic manipulation. However, when arms lack

---

${}^{(1)}\mathbb{1}_{[]}$ is the indicator function: $\mathbb{1}_{[a=b]} = \begin{cases} 1 & \text{if } a = b \\ 0 & \text{otherwise} \end{cases}$

information about competition, they propose a mechanism reliant on the presence of truthful arms, ensuring a substantial revenue for the player.

The most related setting is considered by [8]. They addressed a multi-armed bandit problem with strategic arms, employing a similar utility function, dependent on cumulative savings as outlined in (1). Their work revealed that if the player aims for the highest mean $\mu_1$ – i.e., employs a no-regret strategy – then the arms can form an undesirable equilibrium that leaves the player with a sub-linear cumulative revenue. Consequently, any incentive-unaware learning algorithm that is oblivious to the strategic nature of the arms is not resilient to such a utility definition and will generally fail to achieve low regret. Therefore, it becomes crucial to design algorithms that rely not only on sequential decision-making but also on incorporating incentive mechanisms to enhance truthfulness. In this context, Lemma 14 of [8] establishes that no mechanism can guarantee more than $\mu_2 T$ as revenue for the player in the worst case. This observation justifies the use of $\mu_2$ as a comparator in the regret definition. As a solution, they propose a mechanism referred to here as S-ETC (Strategic Explore Then Commit). When arms have the flexibility to report any value within the range of [0,1], potentially incurring negative utility, S-ETC ensures that the player achieves a revenue of $\mu_2 T$ with sub-linear cumulative regret. However, under debt-free reporting, the player incurs an additional regret of $\mathcal{O}(T^{\frac{2}{3}})$ within a given Nash equilibrium among the arms.

## 2.4 Objectives and Challenges

The central question we address is whether there exist, under debt-free reporting, an algorithm for the player that establishes a Nash equilibrium for the arms, guaranteeing lower regret, akin to classical non-strategic bandit settings [24, 25, 26], or not.

Dealing with strategic arms in the context of debt-free reporting poses significant challenges, especially when striving to minimize regret. Our overarching goal is to achieve a regret of $\mathcal{O}(\frac{\log(T)}{\Delta})$. This task proves to be far from straightforward, as it cannot be resolved by merely adapting prior research. In particular, when considering the fixed design (i.e., non adaptive exploration phase) solution proposed by [8], the minimum bonus required to ensure truthfulness is directly proportional to the length of the exploration phase. This cannot provide a regret guarantee better than $\mathcal{O}(T^{\frac{2}{3}})$.

Furthermore, this observation underscores that achieving incentive-compatibility cannot be solely contingent on the bonus, as it tends to become prohibitively costly for the player. To address this, we shall rely on the adaptive elimination of arms, introducing a secondary trade-off for the arms that enables a reduction in the bonus requirement. The less faithful an arm behaves, the more swiftly it is removed from consideration. However, this approach grants arms some control over the number of their pulls, necessitating meticulous design in other aspects of the mechanism.

## 2.5 Contributions

We operate within the framework of debt-free reporting, wherein each arm is constrained to report values no higher than its observed outcomes. In this context, our contributions are:

1. We introduce a novel incentive-aware learning algorithm, denoted as Algorithm 1: S-SE, which integrates mechanism design and online learning techniques. This algorithm adeptly motivates favorable arm strategies, minimizing regret by adjusting a well-calibrated bonus. This introduces a strategic tradeoff for arms, balancing high savings through dishonesty with the risk of swift elimination.

2. We demonstrate that under Algorithm 1, there is a dominant-strategy SPE in our game, where each arm simply reports truthfully, as proven in Theorem 4.1. Under this equilibrium, the player provably obtain an expected revenue of $\mu_2 T$, up to a sub-linear regret that is bounded by $\mathcal{O}\left(\sum_{k=3}^{K} \frac{\log(T)}{\Delta_{2k}} + \sum_{k=2}^{K} \frac{\log(T)}{\Delta_{1k}}\right)$ where gaps are defined as: $\Delta_{ij} = \mu_i - \mu_j, \forall i, j \in \{1, \ldots, K\}$. In the worst-case scenario with significantly small gaps, the regret bound is of $\mathcal{O}\left(\sqrt{KT \log(T)}\right)$, as outlined in Theorem 4.2. This result outperforms the regret bound of $\mathcal{O}(T^{2/3})$ presented in the literature within the examined setting (see Table 1).

3. Under additional technical assumptions, we analyze the player's regret incurred when employing Algorithm 1 within any arms strategy profile. This analysis aims to provide a comprehensive characterization of a broad range of potential equilibria.

Table 1: Comparison summary between S-ETC [8] and S-SE: setting and results.

| | Tacit Model | Debt-free reporting | Bonus nature | Regret |
|---|---|---|---|---|
| S-ETC [8] | ✓ | ✓ | Additional rounds | $\mathcal{O}(T^{2/3})$ |
| S-SE | ✓ | ✓ | Payment | Problem-dependent $\mathcal{O}(\log(T)/\Delta)$ Problem-independent $\mathcal{O}(\sqrt{KT\log(T)})$ |

## 3 The Player Algorithm

This section is dedicated to the description of the algorithm used by the player. We first introduce some notations.

Let $\{a_{k,t}\}_{k\in\{1,\dots,K\}}$ be a set of $K$ real values evaluated at time $t$. We then denote by $\boldsymbol{a}_t$ the vector obtained by concatenating values $\{a_{k,t}\}_{k\in\{1,\dots,K\}}$, i.e., $\boldsymbol{a}_t^\top = [a_{1,t}, a_{2,t}, \dots, a_{K,t}]$. We also denote by $\sigma_{\boldsymbol{a}}$ a bijection from $\{1, \dots, K\}$ to itself that gives the index of the $k^{\text{th}}$ highest element in vector $\boldsymbol{a}$. For example, if $\boldsymbol{a}^\top = [1,\ 10,\ 4,\ 6]$, then $\sigma_{\boldsymbol{a}}(1)$ represents the index of the highest value in $\boldsymbol{a}$, which is 2. Likewise, $\sigma_{\boldsymbol{a}}(2) = 4$, $\sigma_{\boldsymbol{a}}(3) = 3$, and $\sigma_{\boldsymbol{a}}(4) = 1$. The inverse of this mapping provides the rank associated with a given index. When the context is clear, we can simplify the notation and omit the vector $\boldsymbol{a}$, proceeding directly with $\sigma$ and its inverse $\sigma^{-1}$.

Let $S$ be a set; then we denote the cardinality of $S$ by $|S|$.

### 3.1 Algorithm Presentation

Algorithm 1 closely integrates a strategy that combines successive elimination and bonus allocation, hence it is referred to as **S**trategic **S**uccessive **E**limination (S-SE). It is structured into two distinct phases:

**The Initial Phase.** It is composed of the first $\tau$ rounds (a random stopping time defined in (8)), where an adaptive exploration technique is employed, based on round-robins on active arms leading to a series of successive eliminations aimed at identifying the best-performing arm. To achieve this, the player keeps track of the number of times $n_k(t)$ each arm $k$ is pulled up to time $t$, defined as:

$$n_k(t) = \sum_{s=1}^{t} \mathbb{1}_{[k_s=k]} \tag{3}$$

Additionally, the empirical average of received rewards from arm $k$ at time $t$ is computed as:

$$\tilde{\mu}_{k,t} = \frac{1}{n_k(t)} \sum_{s=1}^{t} x_{k,s} \cdot \mathbb{1}_{[k_s=k]} \tag{4}$$

This is distinct from the empirical average of observed rewards:

$$\hat{\mu}_{k,t} = \frac{1}{n_k(t)} \sum_{s=1}^{t} r_{k,s} \cdot \mathbb{1}_{[k_s=k]} \tag{5}$$

which is observable only by the concerned arm $k$. The relevant quantity to the player is $\tilde{\boldsymbol{\mu}}_t = (\tilde{\mu}_{k,t})_{k\in[K]}$ and its ordering mapping $\sigma_{\tilde{\boldsymbol{\mu}}_t}$. However, to alleviate cumbersome notations, we shall use from now on $\sigma_t$ instead of $\sigma_{\tilde{\boldsymbol{\mu}}_t}$.

Leveraging a Hoeffding confidence bound and denoting $\alpha_{k,t} = \sqrt{\frac{2\log(T)}{n_k(t)}}$, the algorithm effectively eliminates arms that are likely to perform worse than the best arm with high probability. Hence, an arm $k$ is eliminated at time $t$ if the following event is happening:

$$E_k^t := \{\tilde{\mu}_{\sigma_t(1),t} - \alpha_{\sigma_t(1),t} \geq \tilde{\mu}_{k,t} + \alpha_{k,t}\} \tag{6}$$

which leads to define the stopping time at which arm $k$ is eliminated, with the convention that $\inf \emptyset = +\infty$,

$$\tau_k := \inf\{t \in [T] : E_k^t\}. \tag{7}$$

**Algorithm 1:** Strategic Successive Elimination (**S-SE**)

1 Let $\mathcal{A}_1 = \{1, \ldots, K\}$ the set of all arms and t=1
2 **while** $t \leq T$ **and** $|\mathcal{A}_t| > 1$ **do**
3     Play $k \in \{k \in \mathcal{A}_t : \underset{k \in \mathcal{A}_t}{\operatorname{argmin}} \, n_k(t-1)\}$
4     Update $n_k(t)$ and $\tilde{\mu}_{k,t}$
5     **if** $n_k(t) = n_{k'}(t), \, \forall k, k' \in \mathcal{A}_t$ **then**
6       $\mathcal{A}_{t+1} = \mathcal{A}_t \setminus \{k \in \mathcal{A}_t : \tilde{\mu}_{\sigma_t(1),t} - \alpha_{\sigma_t(1),t} \geq \tilde{\mu}_{k,t} + \alpha_{k,t}\}$            ▷ Drop bad arms
7     **else**
8       $\mathcal{A}_{t+1} = \mathcal{A}_t$
9     **end**
10    t=t+1
11 **end**
                                                 ▷ $\tau$ is the length of the first phase
12 Identify the arm with the highest average $\sigma_\tau(1)$, denoted as $k^\star$
13 **for** $t = \tau + 1, \cdots, T$ **do**
14     Play $k^\star$ and receive the reported value $x_{k^\star,t}$
15     Compute $\tilde{x}_{k^\star,t} = \frac{1}{t-\tau} \sum\limits_{s=\tau+1}^{t} x_{k^\star,s}$
16     **if** $\tilde{x}_{k^\star,t} < \min(\tilde{\mu}_{\sigma_\tau(2),\tau}, \tilde{\mu}_{\sigma_\tau(1),\tau} - \sqrt{\frac{2\log(T)}{t-\tau}})$ **then**
17       Set $\Psi_{k^\star} = 0$                                            ▷ Incentive
18       Stop playing $k^\star$ immediately and move to step 21         ▷ Incentive
19     **end**
20 **end**
21 Allocate bonuses $\Psi_{k \in \{1,\ldots,K\}}$                                   ▷ Incentive

Consequently, the number of active arms in the set $\mathcal{A}_t$ progressively reduces over time until, eventually, only the best-performing arm remains. Hence, the duration of this initial phase, represented as $\tau$, is a random stopping time. It is not fixed from the outset but dynamically adapted to the statistical characteristics of the arms, and it is defined as:

$$\tau = \inf\{t \in [T] : |\mathcal{A}_t| = 1\} = \inf\left\{t \in [T] : \exists S_t \subset \{1, \ldots, K\} \text{ s.t } |S_t| = K - 1 \text{ and } \bigcap_{k \in S_t} E_k^t\right\}$$
(8)

**The Second Phase.** It starts after stage $\tau$ and thus may never happen, as in classical MAB. In such case, for $t > \tau$, the best arm, i.e. $k = \sigma_\tau(1)$, is required to report an average value that surpasses at least the second-highest average achieved at the conclusion of the initial phase, denoted as $\tilde{\mu}_{\sigma_\tau(2),\tau}$. If the player confidently detects the best arm is defecting and fails to report as much, it is no longer played. Additionally, its bonus is set to zero, and the game is halted (we assume that the player can choose to end the game when the best arm defects).

At the conclusion of the game, a bonus denoted as $\Psi_k$ is assigned to each arm $k$, based on $\tilde{\boldsymbol{\mu}}_{[\tau]}$ and its rank $\sigma_\tau^{-1}(k)$. The strategic distribution of bonuses serves as an incentive mechanism to encourage truthfulness. Drawing inspiration from real-world practices in financial exchanges within e-commerce, all bonus payments are deferred until the end of the game to ensure incentives.

**Definition 3.1.** *Let $\tau_k$ represent the last time when arm $k$ is active. Let $\tau$ and $\tau'$ respectively denote the duration of the first and second phases. Then the bonus function $\Psi_k$ is defined as following:*

$$\Psi_k(x_k, x_{-k}) = \left(n_k(T)\big(\tilde{\mu}_{k,T} - \tilde{\mu}_{\sigma_{\tau_k-1}(2),\tau_k-1}\big) + x_{k,\tau_k}\right) \mathbb{1}_{[\sigma_\tau(1)=k]} \mathbb{1}_{[\tau+\tau' \geq T]}$$
$$+ \left(\frac{16\log(T)}{\tilde{\mu}_{\sigma_{\tau_k-1}(1),\tau_k-1} - \tilde{\mu}_{k,\tau_k-1}} + x_{k,\tau_k}\right) \mathbb{1}_{[\sigma_\tau^{-1}(k) \geq 2]}$$
(9)

Hence, contingent on their ranking at the conclusion of the first phase:

**The highest-performing arm:** if it refrains from defecting during the second phase, it will be rewarded with a bonus that compensates for the discrepancy between its reported value and the second-best reported value. This process mirrors the dynamics observed in second-price auctions.

**Suboptimal arms:** they will receive a bonus inversely proportional to the difference between their means and the best arm mean. Consequently, the bonus increases as the reported values grow larger.

By implementing anytime tests to eliminate arms during the first phase (line 6 of Algorithm 1), the algorithm introduces a strategic tradeoff for arms, balancing the advantages of dishonesty with the risk of expedited elimination. This tradeoff constrains potential gains from dishonest behavior, resulting in smaller bonuses required to ensure truthful reporting and, consequently, providing a more robust guarantee. In contrast to the fixed design with a predetermined exploration phase length introduced in [8], where arms know from the beginning exactly how long they will be played during the first phase independently of their performances, giving them more freedom and requiring larger bonuses (proportional to the exploration phase) to ensure truthfulness, which is more costly than the bonus given in Definition 3.1.

## 4 Dominant-Strategy SPE

Algorithm 1 combines elements of a bandit algorithm and an incentive mechanism, creating a beneficial tradeoff that encourages truthful reporting as a dominant strategy for each arm in any subgame. Consequently, each arm reporting truthfully forms a dominant-strategy SPE. Specifically, we demonstrate that the utility of arm $k$ under the truthful strategy $\pi_k^\star$ dominates its utility under any other strategy $\pi_k$, given any fixed history[(2)] $h_{k,t} = \{k_z, r_{k_z,z}, x_{k_z,z}\}_{z \leq t:k_z=k} \cup \{k_z\}_{z \leq t}$. Formally, we define the subgame utility of arm $k$ under strategy profile $(\pi_k, \pi_{-k})$, at time $t$ given any history $h_{k,t-1}$ as:

$$\mathcal{U}_k(\pi_k, \pi_{-k})[t : T \mid h_{k,t-1}] = \sum_{s=t}^{T} (r_{k,s} - x_{k,s}) \cdot \mathbb{1}_{[k_s=k]} + \Psi_k(x_k, x_{-k}), \tag{10}$$

where the history $h_{k,t-1}$ is implicitly considered within $\Psi_k(x_k, x_{-k})$.

**Theorem 4.1** (Incentive-Compatibility). *Under Algorithm 1 and using the bonus function in Definition 3.1, for any arm $k$, any strategy $\pi_k$, any strategy profile of other arms $\pi_{-k}$, at any time t, and given any history $h_{k,t-1}$, the truthful reporting strategy $\pi_k^\star$ satisfies:*

$$\mathcal{U}_k(\pi_k, \pi_{-k})[t : T|h_{k,t-1}] \leq \mathcal{U}_k(\pi_k^\star, \pi_{-k})[t : T|h_{k,t-1}] \tag{11}$$

*Proof.* See Appendix C. □

Therefore, truthful reporting is a best response to any strategy profile $\pi_{-k}$. Consequently, each arm playing truthfully forms a dominant-strategy SPE.

**Regret Bound.** Under the dominant truthful SPE established in Theorem 4.1, we compute the player's regret over $T$ rounds. As previously discussed, the regret is calculated in relation to $\mu_2$. Notably, a subtlety arises in the strategic scenario as compared to the non-strategic one: we must factor in the bonuses paid when calculating the regret.

**Theorem 4.2** (Regret Bound). *Let $T \geq 1$. The regret of Algorithm 1 is upper-bounded as:*

$$\mathcal{R}_T \leq \mathcal{O}\left(\sum_{k=2}^{K} \frac{\log(T)}{\Delta_{1k}} + \sum_{k=3}^{K} \frac{\log(T)}{\Delta_{2k}}\right) \tag{12}$$

*The instance-independent regret upper-bound is given by:*

$$\mathcal{R}_T \leq \mathcal{O}\left(\sqrt{KT \log(T)}\right) \tag{13}$$

*Proof.* See Appendix D. □

---

[(2)]By definition, the history combines the information available to the arm to make a decision concerning the reporting, in addition to the reported value, i.e $h_{k_t,t} = \mathcal{F}_{k_t,t} \cup \{x_{k_t,t}\}$.

Theorem 4.2 demonstrates that the regret bound under the considered setting is significantly lower than the regret attained in the Nash equilibrium presented in [8] under the same conditions. The bonus design compensates for any gains that may occur from being untruthful, establishing the truthful SPE. Under the latter, the algorithm is the classical successive elimination algorithm designed for the non-strategic MAB, whose worst-case regret is known to match that of Theorem 4.2. At the same time, the successive eliminations create a trade-off between high savings from dishonesty and swift elimination, which reduces the bonus values needed in a way that they do not incur an additional harmful cost for the regret.

A second observation is that the instance-dependent regret matches the one of a regular MAB $\mathcal{O}\left(\sum_{k=2}^{K} \frac{\log(T)}{\Delta_{1k}}\right)$ up to a second additive term of the same order $\mathcal{O}\left(\sum_{k=3}^{K} \frac{\log(T)}{\Delta_{2k}}\right)$. This second term comes from the additional need to correctly estimate the value of the second-best arm to make sure to "bill" accurately the best arm during the second phase. The less accurate the estimation of $\mu_2$, the larger the deviation the best arm could have in the second phase without being detected. As the gain from potential deviation is incorporated in the bonus to ensure truthfulness, the accuracy of the estimation of $\mu_2$ impacts the regret.

It's worth noting that the truthful SPE highlighted in Theorem 4.1 might raise questions about the regret comparator $\mu_2$. In other words, if truthfulness exists, why can't we expect to achieve rewards of $\mu_1$? While it's true that arms are incentivized to be truthful, this motivation is realized through the allocation of bonuses at the conclusion. Bonuses are part of the arm's utility and contribute to making truthful reporting the best response that maximizes utility. Specifically, the best arm is incentivized with a bonus of $\mathcal{O}(T\Delta_{12})$. Since the bonus is integral to the regret calculation, we compute the regret relative to $\mu_2$ in a manner similar to the approach justified in Lemma 14 of [8].

**Tightness of Regret.** For simplicity, let's consider three arms with $\mu_1 \geq \mu_2 \geq \mu_3$. The optimal arm only needs to report a slightly higher value than the second-best arm to be chosen by the player as the superior option. Consequently, all efficient low-regret mechanisms must ensure truthfulness, at least for the two best arms. Under debt-free reporting, where no arm can falsely claim to be better than it actually is, distinguishing between the two best arms incurs a regret of $\frac{\log(T)}{\Delta_{12}}$. Additionally, distinguishing between the second and the third arm is required to estimate the second-best mean accurately, resulting in a regret of $\frac{\log(T)}{\Delta_{23}}$. However, the third arm does not need to be truthful; we only need to distinguish it from the second-best arm. Therefore, in general, the incurred regret will be of order $\frac{\log(T)}{\tilde{\Delta}_{23}}$, where $\tilde{\Delta}_{23}$ is the difference between the true mean $\mu_2$ and an effective mean $\tilde{\mu}_3$, which can be different from $\mu_3$. Ideally, $\tilde{\mu}_3 = 0$ to minimize the corresponding regret i.e incentivizing non-truthful strategies for arms $k \geq 3$, allowing the player to identify the second-best arm more quickly. Nonetheless, in all cases, the regret will be of the same order as the one described in (12). In Appendix F, we conducted an experimental analysis on simulated data, highlighting the tightness of the regret bounds by evaluating several strategies.

**Utilities Bounds.** Considering the game-like dynamics between the player and the arms, alongside the regret bounds, it is natural to examine bounds on the utilities of the arms, as illustrated in the following corollary.

**Corollary 4.1** (Utilities Upper Bound). *Under the truthful SPE, the utility of any arm is upper bounded as follows:*

$$\forall k \in \{2, \dots, K\} \quad \mathbb{E}[\mathcal{U}_k] \leq \mathcal{O}\left(\frac{\log(T)}{\Delta_{1k}}\right) \quad and \quad \mathbb{E}[\mathcal{U}_1] = \mathcal{O}(T\Delta_{12}) \tag{14}$$

*Proof.* See Appendix D.1. □

## 5 Regret Analysis Across Arbitrary Strategy Profiles

In the previous section, we assessed the regret of Algorithm 1 when the arms adhere to the dominant strategy SPE. General profiles and equilibria in extensive games are known to be intractable [27, 28]. Specifically, for an arbitrary strategy profile that is not truthful, the values reported by the arms are not i.i.d., complicating the analysis as tools like Hoeffding's inequality cannot be directly applied. Consequently, determining the algorithm's output becomes non-trivial. To address this, we introduce an additional technical assumption regarding the boundedness of gains through dishonesty. With

this assumption, we can analyze the algorithm under a wide range of strategy profiles, not just the dominant one previously discussed.

Fix an arbitrary strategy profile $\boldsymbol{\pi}$. For each arm $k$, we define $S_k^T = \sum_{s=1}^T (r_{k_s,s} - x_{k_s,s}) \cdot \mathbb{1}_{[k_s=k]}$ as the cumulative savings of arm $k$ up to round $T$. We assume the existence of some upper-bound $M$ on cumulative saving, i.e., such that for all arms $k \in [K]$ and for any strategy $\boldsymbol{\pi}$, it must hold that $S_k^T \leq M$. We define the effective mean under strategy $\boldsymbol{\pi}$ as $\mu_k^{\boldsymbol{\pi}}$, and let $\boldsymbol{\mu}^{\boldsymbol{\pi}} = (\mu_k^{\boldsymbol{\pi}})_{k \in [K]}$. Therefore, we define $\Delta_k^{\boldsymbol{\pi}}$ as the difference between the highest effective mean under strategy $\boldsymbol{\pi}$ and the effective mean of arm $k$, given by : $\Delta_k^{\boldsymbol{\pi}} = \mu_{\sigma_{\boldsymbol{\mu}^{\boldsymbol{\pi}}}(1)}^{\boldsymbol{\pi}} - \mu_k^{\boldsymbol{\pi}}$. Similarly, $\underline{\Delta}_k^{\boldsymbol{\pi}}$ represents the difference between the second-highest effective mean and the effective mean of arm $k$ under strategy $\boldsymbol{\pi}$, defined as: $\underline{\Delta}_k^{\boldsymbol{\pi}} = \mu_{\sigma_{\boldsymbol{\mu}^{\boldsymbol{\pi}}}(2)}^{\boldsymbol{\pi}} - \mu_k^{\boldsymbol{\pi}}$. We show that when arms use the strategy profile $\boldsymbol{\pi}$, even under non-i.i.d. reported variables, Algorithm 1 outputs the second highest effective mean, and the regret is provided by the following theorem.

**Theorem 5.1.** *For any arbitrary strategy profile $\boldsymbol{\pi}$ with $M$-bounded savings, the regret of Algorithm 1 is bounded by:*

$$\mathcal{R}_T^{\boldsymbol{\pi}} = \mathbb{E}\left[\sum_{t=1}^T \left(\mu_{\sigma_{\boldsymbol{\mu}^{\boldsymbol{\pi}}}(2)}^{\boldsymbol{\pi}} - x_{k_t,t}\right) + \sum_{k=1}^K \Psi_k(x_k, x_{-k})\right]$$

$$\leq \mathcal{O}\left(\sum_{k:\sigma_{\boldsymbol{\mu}^{\boldsymbol{\pi}}}^{-1}(k)\geq 2} \max\left\{M, \frac{\log(T)}{\Delta_k^{\boldsymbol{\pi}}}\right\} + \sum_{k:\sigma_{\boldsymbol{\mu}^{\boldsymbol{\pi}}}^{-1}(k)\geq 3} \max\left\{M, \frac{\log(T)}{\underline{\Delta}_k^{\boldsymbol{\pi}}}\right\}\right) \tag{15}$$

*Proof.* See Appendix E.2. $\square$

The upper bound of savings $M$ determines the regret with regard to the second-highest effective mean of any strategy $\boldsymbol{\pi}$, as shown by Theorem 5.1. In particular, it is demonstrated to be sublinear in $T$ provided that $M = o(T)$. The upper bound exhibits a similar form to the problem-dependent upper bound in Theorem 4.2. Specifically, under the truthful equilibrium $\boldsymbol{\pi}^\star$ where $M = 0$, $\mu_k^{\boldsymbol{\pi}^\star} = \mu_k$, $\Delta_k^{\boldsymbol{\pi}^\star} = \Delta_{1k}$, and $\underline{\Delta}_k^{\boldsymbol{\pi}^\star} = \Delta_{2k}$, we retrieve the result from (12). Additionally, in the debt-free reporting setting, it is observed that for any strategy $\boldsymbol{\pi}$, $\mu_{\sigma_{\boldsymbol{\mu}^{\boldsymbol{\pi}}}(2)}^{\boldsymbol{\pi}} \leq \mu_2$. This suggests that the player's revenue drops when arms diverge from the honest SPE in a way that affects the second-highest mean.

**Sensitivity of Algorithm 1 around truthful SPE.** For a strategy profile where $M = o(\log(T))$, the difference between the true second highest mean $\mu_2$ and the effective second highest mean $\mu_2^{\boldsymbol{\pi}}$ is of the same order as the regret. In this scenario, the player's revenue is $\mu_2 T$, up to a sublinear regret $\mathcal{R}_T^{\boldsymbol{\pi}} = \mathcal{O}(\mathcal{R}_T)$. Hence, even though the savings are not zero, we retrieve results comparable to those with truthful reporting.

## 6 Limitations and Future Work

The algorithm S-SE, presented as an efficient solution to the strategic MAB, combines elements of successive elimination with a well-tuned incentive. This raises the question: is it possible to adapt any no-regret MAB algorithm, such as UCB, with a suitable incentive to solve the same problem? A complete and detailed answer to such a question is reserved for future work. However, our intuition is that this would be challenging because, in a similar setting, [29] argue that a truthful mechanism under the strategic MAB problem needs to be exploration-separated. In other words, at any given time step, either it exploits (action depends on current knowledge) or it explores (action allows to gain knowledge but does not depend on current information), but not both simultaneously. This observation suggests that a tailored version of UCB (at least without additional assumptions) may not be suitable.

Despite the similarity between the algorithm of [8] and our algorithm S-SE, both being strategic versions of exploration-separated algorithms, there is a major difference in the nature of the games generated by both algorithms. In [8], all arms are played sequentially for a predetermined number of explorations each before committing to the best arm, creating a simultaneous-like game. In contrast, Algorithm 1 uses an adaptive design permitting lower regret. However, this adaptability involves an extensive game requiring a more complicated analysis based on the concept of SPE.

It's also worth mentioning that the improvement in regret doesn't depend on the type of incentives, whether they are additional bonus rounds or bonus payments. This is because bonuses are subtracted

from player revenue and considered in the regret calculation, irrespective of their nature. Therefore, replacing the additional rounds in [8] with payments will not lead to an improvement in regret. However, in this paper, actual payments are chosen to keep the intuition more direct and facilitate the presentation.

## 7   Conclusion

We addressed the challenge of strategic multi-armed bandit problems under debt-free reporting. Arms aim to maximize their own utilities by manipulating the reported values to the player, resulting in a more complex problem compared to the standard setting due to the game-like dynamics involved. We proved that by employing a strategic variant of successive elimination algorithm, it is possible to design a bonus-based incentive structure, resulting in a dominant-strategy SPE for arms where they report truthfully. The dynamics of the successive elimination process, particularly the trade-off between high savings and swift elimination, allow for the implementation of cost-effective bonuses, leading to low regret.

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

# A Useful Inequalities

**Theorem A.1** (Hoeffding's Inequality). *Let $X_1, \ldots, X_n$ be a collection of $n$ i.i.d. sampled values from a distribution $\mathcal{D}$ with expected value $\mu$. Define $\hat{\mu} = \frac{1}{n}\sum_{i=1}^{n} X_i$. Then, for any $\epsilon > 0$, we have:*

$$\mathbb{P}\left(|\hat{\mu} - \mu| \geq \epsilon\right) \leq 2\exp\left(-2n\epsilon^2\right). \tag{16}$$

**Fact A.1.** *For sufficiently large $T$, the following inequality holds:*

$$\sum_{t=1}^{\tau}\sqrt{\frac{2\log(T)}{t}} \leq \sqrt{8\log(T)}[\sqrt{\tau} - 1] + 1 \leq \sqrt{8\log(T)}\sqrt{\tau} \tag{17}$$

**Fact A.2.**

$$\sqrt{16\log(T)}\sqrt{\max\left\{\frac{3M}{\Delta_k^\pi}, \frac{162\log(T)}{(\Delta_k^\pi)^2}\right\}} \leq \max\left\{3M, \frac{162\log(T)}{\Delta_k^\pi}\right\} \tag{18}$$

# B Background on Non-Strategic Successive Elimination Algorithm

In the non-strategic case, we assume directly that arms are truthful, i.e., $x_{k,t} = r_{k,t}$ for all $k \in \{1, \ldots, K\}$ and $t \in [T]$ and are independent and identically distributed (i.i.d). The classic successive elimination algorithm is as follows:

---
**Algorithm 2:** Successive Elimination

---
1 Let $\mathcal{A}_1 = \{1, \ldots, K\}$ the set of all arms and t=1
2 **while** $t \leq T$ *and* $|\mathcal{A}_t| > 1$ **do**
3     Play $k \in \{k \in \mathcal{A}_t : \operatorname*{argmin}_{k \in \mathcal{A}_t} n_k(t-1)\}$            ▷ Round-robins instance
4     Update $n_k(t)$ and $\tilde{\mu}_{k,t}$
5     **if** $n_k(t) = n_{k'}(t), \forall k, k' \in \mathcal{A}_t$ **then**
6        $\mathcal{A}_{t+1} = \mathcal{A}_t \setminus \{k \in \mathcal{A}_t : \tilde{\mu}_{\sigma_t(1),t} - \alpha_{\sigma_t(1),t} \geq \tilde{\mu}_{k,t} + \alpha_{k,t}\}$     ▷ Drop bad arms
7     **else**
8        $\mathcal{A}_{t+1} = \mathcal{A}_t$
9     **end**
10    t=t+1
11 **end**
12 Continue to play the remaining best arm $k^\star$

---

During the first phase, the Algorithm 2 plays arms and then eliminates those that are performing poorly, leveraging the corresponding confidence bound $\alpha_{.,.}$. Consequently, the number of active arms in the set $\mathcal{A}_t$ progressively reduces over time until only the best-performing arm remains and is then played until the end of the game. Let $n_k(T)$ be the number of rounds each arm $k$ is played before being eliminated, and let $\tau$ be the total length of the first phase.

**Theorem B.1.** *Suppose that $\Delta_{1k} > 0$, for $k = 2, \ldots, K$. Then under successive elimination algorithm (Algorithm 2):*

- *with probability at least $1 - \frac{4}{T^2}$:*

$$n_k(T) \leq t_k = \frac{32\log(T)}{\Delta_{1k}^2} \tag{19}$$

- *with probability at least $1 - \frac{4K}{T^2}$:*

$$\tau \leq t_1 = \sum_{k=3}^{K}\frac{32\log(T)}{\Delta_{1k}^2} + 2\frac{32\log(T)}{\Delta_{12}^2} \tag{20}$$

*Proof of Theorem B.1.* For a given time instance $t$ and arm $k \in \mathcal{A}_t$, leveraging the Hoeffding's inequality, we can assert the following inequality:

$$\forall t, \quad \mathbb{P}(|\tilde{\mu}_{k,t} - \mu_k| \geq \alpha_{k,t}) \leq \frac{2}{T^4} \tag{21}$$

Employing the union bound, we can deduce that with a probability of at least $1 - \frac{2}{T^2}$, for any time $t$ and arm $k$, the following condition holds:

$$|\tilde{\mu}_{k,t} - \mu_k| \leq \alpha_{k,t} \tag{22}$$

Consequently, with a probability exceeding $1 - \frac{2K}{T^2}$, the elimination of the best arm is avoided.

Consider any arm $k$ such that $\mu_k \leq \mu_1$, i.e., arms that are worse than the best arm. We focus on the last round $\tau_k$ when we did not deactivate $k$ yet. This happens when the first time $\tau_k$, the confidence intervals of $k$ and the best arm do not overlap, i.e the first time when event $E_k^t$ is valid. A suboptimal arm $k$, is played at time $t$ if its upper confidence bound exceeds the lower confidence bound of arm 1, indicating that:

$$\tilde{\mu}_{k,t} + \alpha_{k,t} > \tilde{\mu}_{1,t} - \alpha_{1,t} \tag{23}$$

Using (22), we get:

$$\tilde{\mu}_{1,t} \geq \mu_1 - \alpha_{1,t} \quad \text{and} \quad \tilde{\mu}_{k,t} \leq \mu_k + \alpha_{k,t} \tag{24}$$

Hence with probability at least $1 - \frac{4}{T^2}$:

$$\mu_k + 2\alpha_{k,t} \qquad \geq \mu_1 - 2\alpha_{1,t} \tag{25}$$
$$\Rightarrow \qquad 2\alpha_{k,t} + 2\alpha_{1,t} \qquad \geq \mu_1 - \mu_k \tag{26}$$
$$\Rightarrow \qquad 4\alpha_{k,t} \qquad \geq \mu_1 - \mu_k \qquad (n_k(t) = n_1(t)) \tag{27}$$
$$\Rightarrow \qquad \frac{32\log(T)}{(\mu_1 - \mu_k)^2} \qquad \geq n_k(T) \tag{28}$$

Implying that with probability at least $1 - \frac{4}{T^2}$, $n_k(T) \leq t_k$ such that:

$$t_k = \frac{32\log(T)}{(\mu_1 - \mu_k)^2} = \frac{32\log(T)}{\Delta_{1k}^2} \tag{29}$$

So in this case, with a probability of at least $1 - \frac{4K}{T^2}$, $\tau \leq t_1$ such that:

$$t_1 = \sum_{k=3}^{K} t_k + 2t_2 \tag{30}$$

$\square$

## C   Proof of Theorem 4.1

**Theorem 4.1** (Incentive-Compatibility). *Under Algorithm 1 and using the bonus function in Definition 3.1, for any arm $k$, any strategy $\pi_k$, any strategy profile of other arms $\pi_{-k}$, at any time $t$, and given any history $h_{k,t-1}$, the truthful reporting strategy $\pi_k^\star$ satisfies:*

$$\mathcal{U}_k(\pi_k, \pi_{-k})[t : T|h_{k,t-1}] \leq \mathcal{U}_k(\pi_k^\star, \pi_{-k})[t : T|h_{k,t-1}] \tag{11}$$

*Proof.* Let's give $k \in [K]$, $\pi_k \in \mathcal{P}$ and $\pi_{-k} \in \mathcal{P}^{K-1}$. The subgame utility of arm $k$ when the player is using Algorithm 1 and given any history $h_{k,t-1}$ is

$$\mathcal{U}_k(\pi_k, \pi_{-k})[t : T|h_{k,t-1}] = \sum_{s=t}^{T} (r_{k,s} - x_{k,s}) \cdot \mathbb{1}_{[k_s=k]} + \Psi_k(x_k, x_{-k}) \tag{31}$$

where the history $h_{k,t-1}$ explicitly appears in $\Psi_k(x_k, x_{-k})$. For ease of presentation, for $t' \geq t$, we define the windowed savings average $\bar{S}_k^{t:t'} = \frac{1}{n_k(T)} \sum_{s=t}^{t'} (r_{k,s} - x_{k,s}) \cdot \mathbb{1}_{[k_s=k]}$, which represents the contribution of the time window $[t, t']$ to the complete average of savings by arm $k$.

The idea is to analyze both optimal and suboptimal arms.

**Case 1** $-\exists t \leq T, s.t.\ \tilde{\mu}_{k,t} + \alpha_{k,t} < \tilde{\mu}_{\sigma_t(1),t} - \alpha_{\sigma_t(1),t}$ :

Let $\tau_k$ be the last node or round of the extensive game where arm $k$ is played.

In such case, $\tau_k \leq T$ and

$$\mathcal{U}_k(\boldsymbol{\pi}_k, \boldsymbol{\pi}_{-k})[t:T|h_{k,t-1}] = \sum_{s=t-1}^{T} (r_{k,s} - x_{k,s}) \cdot \mathbb{1}_{[k_s=k]} + \Psi_k(x_k, x_{-k}) \tag{32}$$

$$= \sum_{s=t-1}^{\tau_k} (r_{k,s} - x_{k,s}) \cdot \mathbb{1}_{[k_s=k]} + \Psi_k(x_k, x_{-k}) \tag{33}$$

$$= n_k(\tau_k)\bar{S}_k^{t:\tau_k} + \Psi_k(x_k, x_{-k}) \qquad \text{(def. of } \bar{S}_k^{t-1:\tau_k}\text{)} \tag{34}$$

$$= n_k(\tau_k)\bar{S}_k^{t:\tau_k} + \frac{16\log(T)}{\tilde{\mu}_{\sigma_{\tau_k-1}(1),\tau_k-1} - \tilde{\mu}_{k,\tau_k-1}} + x_{k,\tau_k} \qquad \text{(def. of } \Psi_k\text{)} \tag{35}$$

$$= \frac{2\log(T)}{\alpha_{k,\tau_k}^2}\bar{S}_k^{t:\tau_k} + \frac{16\log(T)}{\tilde{\mu}_{\sigma_{\tau_k-1}(1),\tau_k-1} - \tilde{\mu}_{k,\tau_k-1}} + x_{k,\tau_k} \qquad \text{(def. of } \alpha_{k,\tau_k}\text{)} \tag{36}$$

Considering that the final reported value at $\tau_k$ is fully reimbursed to the arm as a bonus, it is dominant to have $r_{k,\tau_k} = x_{k,\tau_k}$ implying that $\bar{S}_k^{t:\tau_k} \leq \bar{S}_k^{t:\tau_k-1}$. The utility is upper-bounded as follows:

$$\mathcal{U}_k(\boldsymbol{\pi}_k, \boldsymbol{\pi}_{-k})[t:T|h_{k,t-1}] \leq \frac{2\log(T)}{\alpha_{k,\tau_k}^2}\bar{S}_k^{t:\tau_k-1} + \frac{16\log(T)}{\tilde{\mu}_{\sigma_{\tau_k-1}(1),\tau_k-1} - \tilde{\mu}_{k,\tau_k-1}} + r_{k,\tau_k} \tag{37}$$

Then, by definition of $\tau_k$, the following inequalities holds,

$$\frac{\tilde{\mu}_{\sigma_{\tau_k-1}(1),\tau_k-1} - \tilde{\mu}_{k,\tau_k-1}}{2} \leq \alpha_{k,\tau_k-1} = \alpha_{k,\tau_k}\sqrt{\frac{n_k(\tau_k)}{n_k(\tau_k-1)}} \tag{38}$$

$$\leq \alpha_{k,\tau_k}\sqrt{\frac{n_k(\tau_k)}{n_k(\tau_k)-1}} \tag{39}$$

$$\leq \alpha_{k,\tau_k}\sqrt{\frac{1}{1 - \frac{\alpha_{k,\tau_k}^2}{2\log(T)}}} \tag{40}$$

By rearranging terms, we obtain:

$$n_k(\tau_k) = \frac{2\log(T)}{\alpha_{k,\tau_k}^2} \leq \frac{8\log(T)}{(\tilde{\mu}_{\sigma_{\tau_k-1}(1),\tau_k-1} - \tilde{\mu}_{k,\tau_k-1})^2} + 1 \tag{41}$$

Hence:

$$\mathcal{U}_k(\boldsymbol{\pi}_k, \boldsymbol{\pi}_{-k})[t:T|h_{k,t-1}] \leq \left(\frac{8\log(T)}{(\tilde{\mu}_{\sigma_{\tau_k-1}(1),\tau_k-1} - \tilde{\mu}_{k,\tau_k-1})^2} + 1\right)\bar{S}_k^{t:\tau_k-1} + \frac{16\log(T)}{\tilde{\mu}_{\sigma_{\tau_k-1}(1),\tau_k-1} - \tilde{\mu}_{k,\tau_k-1}} + r_{k,\tau_k} \tag{42}$$

$$\leq \frac{16\log(T)}{(\tilde{\mu}_{\sigma_{\tau_k-1}(1),\tau_k-1} - \tilde{\mu}_{k,\tau_k-1})^2}\bar{S}_k^{t:\tau_k-1} + \frac{16\log(T)}{\tilde{\mu}_{\sigma_{\tau_k-1}(1),\tau_k-1} - \tilde{\mu}_{k,\tau_k-1}} + r_{k,\tau_k} \tag{43}$$

$$\leq \underbrace{\frac{16\log(T)}{(\tilde{\mu}_{\sigma_{\tau_k-1}(1),\tau_k-1} - \hat{\mu}_{k,\tau_k-1} + \bar{S}_k^{t:\tau_k-1} + \bar{S}_k^{1:t-1})^2}\bar{S}_k^{t:\tau_k-1} + \frac{16\log(T)}{\tilde{\mu}_{\sigma_{\tau_k-1}(1),\tau_k} - \hat{\mu}_{k,\tau_k-1} + \bar{S}_k^{t:\tau_k-1} + \bar{S}_k^{1:t-1}} + r_{k,\tau_k}}_{\zeta(\bar{S}_k^{t:\tau_k-1})} \tag{44}$$

$$\leq \max_{\epsilon \in \mathbb{R}_+} \zeta(\epsilon) = \zeta(0) \tag{45}$$

$$\leq \mathcal{U}_k(\boldsymbol{\pi}_k^\star, \boldsymbol{\pi}_{-k})[t:T|h_{k,t-1}] \tag{46}$$

This concludes the first case.

**Case 2** – $\exists \tau \leq T, s.t.\ \tilde{\mu}_{k,\tau} - \alpha_{k,\tau} > \tilde{\mu}_{\sigma_\tau(2),\tau} + \alpha_{\sigma_\tau(2),\tau}$. In such a case, $\tau_k = \tau + \tau'$, where $\tau$ is the length of the first period, and $\tau'$ is the length of the second one.

$$\mathcal{U}_k(\boldsymbol{\pi}_k, \boldsymbol{\pi}_{-k})[t:T|h_{k,t-1}] = \sum_{t=1}^{\tau+\tau'} (r_{k,t} - x_{k,t}) \cdot \mathbb{1}_{[k_t=k]} + \Psi_k(x_k, x_{-k})\mathbb{1}_{[\tau'=T-\tau]} \quad \text{(Step 17 of Algorithm 1)}$$
$$\tag{47}$$

$$= n_k(T)\bar{S}_k^{t:T} + \Psi_k(x_k, x_{-k})\mathbb{1}_{[\tau'=T-\tau]} \quad \text{(def. of } \bar{S}_k^{t:T}) \tag{48}$$

$$= n_k(T)\bar{S}_k^{t:T} + \left(n_k(T)(\tilde{\mu}_{k,T} - \tilde{\mu}_{\sigma_{\tau-1}(2),\tau-1}) + x_{k,\tau}\right)\mathbb{1}_{[\tau'=T-\tau]} \quad \text{(def. of } \Psi_k) \tag{49}$$

$$= n_k(T)\bar{S}_k^{t:T} + \left(n_k(T)(\hat{\mu}_{k,T} - \bar{S}_k^{1:T} - \tilde{\mu}_{\sigma_{\tau-1}(2),\tau-1}) + x_{k,\tau}\right)\mathbb{1}_{[\tau'=T-\tau]} \tag{50}$$

$$\tag{51}$$

**If** $t > \tau$, meaning that $t$ is in the second phase:

$$\mathcal{U}_k(\boldsymbol{\pi}_k, \boldsymbol{\pi}_{-k})[t:T|h_{k,t-1}] = n_k(T)\bar{S}_k^{t:T} + \left(n_k(T)(\hat{\mu}_{k,T} - \bar{S}_k^{1:T} - \tilde{\mu}_{\sigma_{\tau-1}(2),\tau-1}) + x_{k,\tau}\right)\mathbb{1}_{[\tau'=T-\tau]}$$
$$\tag{52}$$

$$= n_k(T)\bar{S}_k^{t:T} + \left(n_k(T)(\hat{\mu}_{k,T} - \bar{S}_k^{1:t-1} - \bar{S}_k^{t:T} - \tilde{\mu}_{\sigma_{\tau-1}(2),\tau-1}) + x_{k,\tau}\right)\mathbb{1}_{[\tau'=T-\tau]} \tag{53}$$

$$\tag{54}$$

Then, we obtain three cases that depend on $\bar{S}_k^{t:T}$:

- **If** $\bar{S}_k^{t:T} = 0$:
  $$\mathcal{U}_k(\boldsymbol{\pi}_k, \boldsymbol{\pi}_{-k})[t:T|h_{k,t-1}] = n_k(T)(\hat{\mu}_{k,T} - \bar{S}_k^{1:t-1} - \tilde{\mu}_{\sigma_{\tau-1}(2),\tau-1}) + x_{k,\tau} \tag{55}$$
  with $n_k(T) = \mathcal{O}(T)$

- **If** $\bar{S}_k^{t:T} \neq 0$ but still not significant to be detected, i.e $\tau' = T - \tau$:
  $$\mathcal{U}_k(\boldsymbol{\pi}_k, \boldsymbol{\pi}_{-k})[t:T|h_{k,t-1}] = n_k(T)\bar{S}_k^{t:T} + n_k(T)(\hat{\mu}_{k,T} - \bar{S}_k^{1:t-1} - \bar{S}_k^{t:T} - \tilde{\mu}_{\sigma_{\tau-1}(2),\tau-1}) + x_{k,\tau}$$
  $$\tag{56}$$
  $$\leq n_k(T)(\hat{\mu}_{k,T} - \bar{S}_k^{1:t-1} - \tilde{\mu}_{\sigma_{\tau-1}(2),\tau-1}) + x_{k,\tau} \tag{57}$$
  $$\leq \mathcal{U}_k(\boldsymbol{\pi}_k^\star, \boldsymbol{\pi}_{-k})[t:T|h_{k,t-1}] \tag{58}$$

- **If** $\bar{S}_k^{t:T} \neq 0$ in a significant way such that the arm is detected as untruthful, stopped from being played, and prevented from receiving the bonus. In this case at most $\tau' = \frac{2\log(T)}{\alpha_{k,\tau}^2}$:
  $$\mathcal{U}_k(\boldsymbol{\pi}_k, \boldsymbol{\pi}_{-k})[t:T|h_{k,t-1}] = n_k(T)\bar{S}_k^{t:T} \tag{59}$$
  $$\leq \mathcal{U}_k(\boldsymbol{\pi}_k^\star, \boldsymbol{\pi}_{-k})[t:T|h_{k,t-1}] \quad \left(\text{because in this case } n_k(T) = \mathcal{O}(\frac{2\log(T)}{\Delta^2}) = o(T)\right)$$
  $$\tag{60}$$

Hence, it is dominant to report truthfully:
$$\mathcal{U}_k(\boldsymbol{\pi}_k, \boldsymbol{\pi}_{-k})[t:T|h_{k,t-1}] \leq \mathcal{U}_k(\boldsymbol{\pi}_k^\star, \boldsymbol{\pi}_{-k})[t:T|h_{k,t-1}] \tag{61}$$

**If** $t \leq \tau$, meaning that $t$ is in the first phase:

$$\mathcal{U}_k(\boldsymbol{\pi}_k, \boldsymbol{\pi}_{-k})[t:T|h_{k,t-1}] = n_k(T)\bar{S}_k^{t:T} + \left(n_k(T)(\hat{\mu}_{k,T} - \bar{S}_k^{1:T} - \tilde{\mu}_{\sigma_{\tau-1}(2),\tau-1}) + x_{k,\tau}\right)\mathbb{1}_{[\tau'=T-\tau]}$$
$$\tag{62}$$

$$= n_k(T)(\bar{S}_k^{t:\tau} + \bar{S}_k^{\tau+1:T}) + \left(n_k(T)(\hat{\mu}_{k,T} - \bar{S}_k^{1:t-1} - \bar{S}_k^{t:\tau} - \bar{S}_k^{\tau+1:T} - \tilde{\mu}_{\sigma_{\tau-1}(2),\tau-1}) + x_{k,\tau}\right)\mathbb{1}_{[\tau'=T-\tau]}$$
$$\tag{63}$$

Using results from the previous part:
$$\mathcal{U}_k(\boldsymbol{\pi}_k, \boldsymbol{\pi}_{-k})[t:T|h_{k,t-1}] \leq n_k(T)\bar{S}_k^{t:\tau} + n_k(T)(\hat{\mu}_{k,T} - \bar{S}_k^{1:t-1} - \bar{S}_k^{t:\tau} - \tilde{\mu}_{\sigma_{\tau-1}(2),\tau-1}) + x_{k,\tau} \tag{64}$$
$$\leq n_k(T)(\hat{\mu}_{k,T} - \bar{S}_k^{1:t-1} - \tilde{\mu}_{\sigma_{\tau-1}(2),\tau-1}) + r_{k,\tau} \tag{65}$$
$$\leq \mathcal{U}_k(\boldsymbol{\pi}_k^\star, \boldsymbol{\pi}_{-k})[t:T|h_{k,t-1}] \tag{66}$$
This concludes the second case.

**Case 3** – $\forall t \leq T$, $\tilde{\mu}_{k,t} + \alpha_{k,t} \geq \tilde{\mu}_{\sigma_t(1),t} - \alpha_{\sigma_t(1),t}$ **and** $\tilde{\mu}_{k,t} - \alpha_{k,t} \leq \tilde{\mu}_{\sigma_t(2),t} + \alpha_{\sigma_t(2),t}$. In such a case, $\tau$, which represents the length of the first phase, is equal to $T$, meaning that there is no exploitation phase. The utility is:

$$\mathcal{U}_k(\boldsymbol{\pi}_k, \boldsymbol{\pi}_{-k})[t:T|h_{k,t-1}] = n_k(T)\bar{S}_k^{t:T} + \Psi_k(x_k, x_{-k}) \tag{67}$$

$$\leq n_k(T)\bar{S}_k^{t:T} + \left(n_k(T)\big(\tilde{\mu}_{k,T} - \tilde{\mu}_{\sigma_{\tau_k-1}(2),T-1}\big) + x_{k,T}\right) \mathbb{1}_{[\sigma_T^{-1}(k)=1]} \tag{68}$$

$$+ \left(\frac{16\log(T)}{\tilde{\mu}_{\sigma_{T-1}(1),T-1} - \tilde{\mu}_{k,T-1}} + x_{k,T}\right) \mathbb{1}_{[\sigma_T^{-1}(k)\geq 2]} \tag{69}$$

$$\leq \left(n_k(T)\bar{S}_k^{t:T} + n_k(T)\big(\tilde{\mu}_{k,T} - \tilde{\mu}_{\sigma_{T-1}(2),T-1}\big) + r_{k,T}\right) \mathbb{1}_{[\sigma_T^{-1}(k)=1]} \quad \text{(As in case 2)} \tag{70}$$

$$+ \left(n_k(T)\bar{S}_k^{t:T} + \frac{16\log(T)}{\tilde{\mu}_{\sigma_{T-1}(1),T-1} - \tilde{\mu}_{k,T-1}} + r_{k,T}\right) \mathbb{1}_{[\sigma_T^{-1}(k)\geq 2]} \quad \text{(As in case 1)} \tag{71}$$

Hence, we retrieve the two previous cases, and we can conclude that:

$$\mathcal{U}_k(\boldsymbol{\pi}_k, \boldsymbol{\pi}_{-k})[t:T|h_{k,t-1}] \leq \mathcal{U}_k(\boldsymbol{\pi}_k^\star, \boldsymbol{\pi}_{-k})[t:T|h_{k,t-1}] \tag{72}$$

$\square$

# D  Proof of Theorem 4.2

**Theorem 4.2** (Regret Bound). *Let $T \geq 1$. The regret of Algorithm 1 is upper-bounded as:*

$$\mathcal{R}_T \leq \mathcal{O}\left(\sum_{k=2}^{K} \frac{\log(T)}{\Delta_{1k}} + \sum_{k=3}^{K} \frac{\log(T)}{\Delta_{2k}}\right) \tag{12}$$

*The instance-independent regret upper-bound is given by:*

$$\mathcal{R}_T \leq \mathcal{O}\left(\sqrt{KT\log(T)}\right) \tag{13}$$

*Proof.* Theorem 4.1 demonstrates the incentivized truthfulness of the arms. Consequently, we calculate the regret while operating under the assumption that arms adhere to truthfulness, accurately reporting their real values. As previously discussed, the regret is calculated in relation to $\mu_2$. Notably, a subtlety arises in the strategic scenario as compared to the non-strategic one: we must factor in the bonuses paid when calculating the regret. We commence our analysis by closely examining the Bonus function, denoted as $\Psi_k$:

$$\Psi_k(x_k, x_{-k}) = \left(n_k(T)\big(\tilde{\mu}_{k,T} - \tilde{\mu}_{\sigma_{\tau_k-1}(2),\tau_k-1}\big) + x_{k,\tau_k}\right) \mathbb{1}_{[\sigma_\tau(1)=k]} \mathbb{1}_{[\tau+\tau'\geq T]}$$

$$+ \left(\frac{16\log(T)}{\tilde{\mu}_{\sigma_{\tau_k-1}(1),\tau_k-1} - \tilde{\mu}_{k,\tau_k-1}} + x_{k,\tau_k}\right) \mathbb{1}_{[\sigma_\tau^{-1}(k)\geq 2]} \tag{73}$$

- If $\sigma_\tau^{-1}(k) = 1$, then $\Psi_k(x_k, x_{-k}) \leq n_k(T)\big(\tilde{\mu}_{k,T} - \tilde{\mu}_{\sigma_{\tau_k-1}(2),\tau_k-1}\big) + r_{k,\tau_k}$ and given that regret is computed with respect to the second-highest mean then this bonus contributes to the regret only with $r_{k,\tau_k} \leq 1$.

- If $\sigma_\tau^{-1}(k) \geq 2$, then:

$$\Psi_k(x_k, x_{-k}) = \frac{16\log(T)}{\tilde{\mu}_{\sigma_{\tau_k-1}(1),\tau_k-1} - \tilde{\mu}_{k,\tau_k-1}} + r_{k,\tau_k} \leq \overbrace{2\tau_k\big(\tilde{\mu}_{\sigma_{\tau_k-1}(1),\tau_k-1} - \tilde{\mu}_{k,\tau_k-1}\big)}^{(b)} + 1 \tag{74}$$

Interestingly, the term (b) of the upper-bound is directly linked to the regret associated with selecting this arm during the initial phase. Equivalently, we can substitute (b) contribution by considering that this arm contributes with an additional regret, on average per round, $2\Delta_{1k}$ throughout the first phase.

Then the regret is as follows:

$$\mathcal{R}_T \leq \overbrace{\sum_{k=3}^{K} \Delta_{2k}\mathbb{E}[n_k(T)]}^{\text{Exploration regret denoted as } \mathcal{R}_T^1} + \overbrace{\mathbb{E}\left[(T-\tau)(\mu_2 - \tilde{\mu}_{\sigma_\tau(2),\tau})\right] + \mathbb{E}\left[\sum_{t=1}^{\tau} \sqrt{\frac{2\log(T)}{t}}\right]}^{\text{Exploitation regret denoted as } \mathcal{R}_T^2} + \overbrace{\sum_{k=2}^{K} 2\Delta_{1k}\mathbb{E}\left[n_k(T)\right] + K}^{\text{Bonus regret denoted as } \mathcal{R}_T^3}$$

$$\tag{75}$$

Yet, we will evaluate each term of the regret separately:

**Exploration regret $\mathcal{R}_T^1$:**

$$\mathcal{R}_T^1 = \sum_{k=3}^{K} \Delta_{2k}\mathbb{E}[n_k(T)] \tag{76}$$

$$\leq \sum_{k=3}^{K} \left(\Delta_{2k}t_k\mathbb{P}(n_k(T) \leq t_k) + \mu_2 T\mathbb{P}(n_k(T) \geq t_k)\right) \tag{77}$$

$$\overset{(29)}{\leq} \sum_{k=3}^{K} \left(\Delta_{2k}\frac{32\log(T)}{\Delta_{1k}^2} + \mu_2 T\frac{4}{T^2}\right) \qquad \text{(Theorem B.1)} \tag{78}$$

$$\leq \sum_{k=3}^{K} \frac{32\log(T)}{\Delta_{2k}} + o(1) \tag{79}$$

The final inequality is a result of $\Delta_{2k} \leq \Delta_{1k}$.

**Exploitation regret $\mathcal{R}_T^2$:** Using Fact A.1,

$$\mathbb{E}\left[\sum_{t=1}^{\tau} \sqrt{\frac{2\log(T)}{t}}\right] \leq \sqrt{8\log(T)}\mathbb{E}[\sqrt{\tau}] \tag{80}$$

$$\leq \sqrt{8\log(T)}\left[\sqrt{t_1}\mathbb{P}(\tau \leq t_1) + T\mathbb{P}(\tau \geq t_1)\right] \tag{81}$$

$$\leq \sqrt{8\log(T)}\sqrt{t_1} + o(1) \tag{82}$$

$$\leq \sqrt{8\log(T)}\sqrt{2\sum_{k=2}^{K} t_k} + o(1) \tag{83}$$

$$\leq \sqrt{16\sum_{k=2}^{K} \frac{32\log(T)^2}{\Delta_{1k}^2}} + o(1) \tag{84}$$

$$\leq 16\sum_{k=2}^{K} \frac{\sqrt{2}\log(T)}{\Delta_{1k}} + o(1) \tag{85}$$

On the other hand:

$$\mathbb{E}\left[(T-\tau)(\mu_2 - \tilde{\mu}_{\sigma_\tau(2),\tau})\right] \leq T\,\mathbb{E}\left[\mu_2 - \tilde{\mu}_{\sigma_\tau(2),\tau}\right] \tag{86}$$

$$\leq \mathbb{E}\left[\mu_2 - \tilde{\mu}_{\sigma_\tau(2),\tau}|\tau \leq t_1\right]T\mathbb{P}(\tau \leq t_1) + \mu_2 T\mathbb{P}(\tau \geq t_1) \tag{87}$$

$$\leq \mathbb{E}\left[\mu_2 - \tilde{\mu}_{\sigma_\tau(2),\tau}|\tau \leq t_1\right]T + o(1) \tag{88}$$

However, we have:

$$\mu_2 = \mathbb{E}\left[\tilde{\mu}_{\sigma_\tau(2),\tau}|\tau \leq t_1\right]\mathbb{P}(\tau \leq t_1) + \mathbb{E}\left[\tilde{\mu}_{\sigma_\tau(2),\tau}|\tau \geq t_1\right]\mathbb{P}(\tau \geq t_1) \tag{89}$$

$$\Rightarrow \quad \mathbb{E}\left[\tilde{\mu}_{\sigma_\tau(2),\tau}|\tau \leq t_1\right] = \frac{\mu_2 - \mathbb{E}\left[\tilde{\mu}_{\sigma_\tau(2),\tau}|\tau \geq t_2\right]\mathbb{P}(\tau \geq t_2)}{\mathbb{P}(\tau \leq t_1)} \tag{90}$$

$$\geq \frac{\mu_2 - \frac{2K}{T^2}}{1} \tag{91}$$

$$\Rightarrow T\mathbb{E}\left[\mu_2 - \tilde{\mu}_{\sigma_\tau(2),\tau}|\tau \leq t_1\right] \leq \frac{2K}{T} = o(1) \tag{92}$$

Hence:

$$\mathcal{R}_T^2 \leq 16 \sum_{k=2}^{K} \frac{\sqrt{2} \log(T)}{\Delta_{1k}} + o(1) \tag{93}$$

**Bonus regret $\mathcal{R}_T^3$:**

$$\mathcal{R}_T^3 = \sum_{k=2}^{K} 2\Delta_{1k} \mathbb{E}\left[n_k(T)\right] + K \tag{94}$$

$$\leq \sum_{k=2}^{K} 2\Delta_{1k} t_k \mathbb{P}(n_k(T) \leq t_k) + 2T\mathbb{P}(n_k(T) \geq t_k) + K \tag{95}$$

$$\overset{(29)}{\leq} \sum_{k=2}^{K} \left( 2\Delta_{1k} \frac{32 \log(T)}{\Delta_{1k}^2} + 2T \frac{4}{T^2} \right) + K \qquad \text{(Theorem B.1)} \tag{96}$$

$$\leq \sum_{k=2}^{K} \frac{64 \log(T)}{\Delta_{1k}} + K + o(1) \tag{97}$$

By considering (79), (93) and (97) then with a high probability exceeding $1 - \mathcal{O}\left(\frac{1}{T^2}\right)$, the total regret is bounded by:

$$\mathcal{R}_T \leq \mathcal{O}\left( \sum_{k=3}^{K} \frac{\log(T)}{\Delta_{2k}} + \sum_{k=2}^{K} \frac{\log(T)}{\Delta_{1k}} \right) \tag{98}$$

Further analysis can be conducted to eliminate the gaps in the aforementioned regret. To address the variable $\Delta_{ij}$, let us select a fixed $\epsilon > 0$ and proceed as follows:

- Arms $k$ where $\Delta_{ij} \leq \epsilon$ contribute a maximum of $\epsilon$ per round, yielding a cumulative contribution of $\epsilon T$.

- Arms $k$ where $\Delta_{ij} \geq \epsilon$ contribute at most $\mathcal{O}\left(\frac{\log(T)}{\epsilon}\right)$.

By combining these points, the following can be established:

$$\mathcal{O}\left( \sum_{k=3}^{K} \frac{\log(T)}{\Delta_{2k}} + \sum_{k=2}^{K} \frac{\log(T)}{\Delta_{1k}} \right) = \mathcal{O}\left( \frac{K \log(T)}{\epsilon} + \epsilon T \right) \overset{\epsilon = \sqrt{\frac{K \log(T)}{T}}}{\leq} \mathcal{O}(\sqrt{KT \log(T)}) \tag{99}$$

Consequently:

$$\mathcal{R}_T \leq \mathcal{O}\left( \sqrt{KT \log(T)} \right) \tag{100}$$

This analysis serves to bridge the gaps in the regret computation, leading to a more refined understanding of the upper bound on the regret. $\square$

## D.1 Proof of Corollary 4.1

**Corollary 4.1** (Utilities Upper Bound). *Under the truthful SPE, the utility of any arm is upper bounded as follows:*

$$\forall k \in \{2, \ldots, K\} \quad \mathbb{E}[\mathcal{U}_k] \leq \mathcal{O}\left( \frac{\log(T)}{\Delta_{1k}} \right) \quad \text{and} \quad \mathbb{E}[\mathcal{U}_1] = \mathcal{O}(T\Delta_{12}) \tag{14}$$

*Proof.* For the suboptimal arms: $k \in \{2, \ldots, K\}$, using 74 we have:

$$\mathbb{E}\left[\Psi_k\right] \leq 2\Delta_{1k}\mathbb{E}\left[n_k(T)\right] + 1 \tag{101}$$

$$\leq 2\Delta_{1k}t_k\mathbb{P}\Big(n_k(T) \leq t_k\Big) + 2T\mathbb{P}\Big(n_k(T) \geq t_k\Big) + 1 \tag{102}$$

$$\overset{(29)}{\leq} 2\Delta_{1k}\frac{32\log(T)}{\Delta_{1k}^2} + 2T\frac{4}{T^2} + 1 \qquad \text{(Theorem B.1)} \tag{103}$$

$$\leq \mathcal{O}\left(\frac{\log(T)}{\Delta_{1k}}\right) \tag{104}$$

For the optimal arm $k = 1$, the result directly follows from the definition of the corresponding bonus. $\qquad\square$

# E   Proof of Theorem 5.1

## E.1   Number of Pulls under Strategies with Upper-Bounded Savings

**Lemma E.1.** *For any strategy $\boldsymbol{\pi}$ with bounded savings by $M$, the expected number of times that a suboptimal arm $k$ is pulled up to time $T$ can be bounded as follow:*

$$\mathbb{E}[n_k(T)] \leq \max\left\{\frac{6M}{\Delta_k^{\boldsymbol{\pi}}}, \frac{162\log(T)}{(\Delta_k^{\boldsymbol{\pi}})^2}\right\} + \frac{2}{T^2} \tag{105}$$

*Proof.* For simplicity, we omit the dependency on the strategy profile $\boldsymbol{\pi}$ in our notation, since it is clear that we focus on an arbitrary profile $\boldsymbol{\pi}$. Let $\zeta_k(T) = \max\left\{\frac{6M}{\Delta_k^{\boldsymbol{\pi}}}, \frac{162\log(T)}{(\Delta_k^{\boldsymbol{\pi}})^2}\right\}$ be a threshold on the number of pulls of a suboptimal arm $k$.

$$\mathbb{E}\left[n_k(T)\right] \leq \mathbb{E}\left[\sum_{t=1}^{T}\mathbb{1}_{[k_t=k]}\right] \tag{106}$$

$$\leq \mathbb{E}\left[\sum_{t=1}^{T}\mathbb{1}_{[k_t=k,n_k(t-1)\leq\zeta_k(T)]}\right] + \mathbb{E}\left[\sum_{t=1}^{T}\mathbb{1}_{[k_t=k,n_k(t-1)\geq\zeta_k(T)]}\right] \tag{107}$$

$$\leq \zeta_k(T) + \sum_{t=1}^{T}\mathbb{P}\Big(k_t = k, n_k(t-1) \geq \zeta_k(T)\Big) \tag{108}$$

Next, we bound the probability $\mathbb{P}\Big(k_t = k, n_k(t-1) \geq \zeta_k(T)\Big)$ using a union bound. We denote by $k^\star$ the best arm under policy $\boldsymbol{\pi}$. We recall that the elimination test is conducted after arms have been played the same number of times. This means that if we assess whether an arm $k$ is played at time $t$,

then we have $\alpha_{k^\star,t-1} = \alpha_{k,t-1}$.

$$\mathbb{P}\Big(k_t = k, n_k(t-1) \geq \zeta_k(T)\Big) = \mathbb{P}\Big(\tilde{\mu}_{k,t-1} + \alpha_{k,t-1} \geq \tilde{\mu}_{k^\star,t-1} - \alpha_{k^\star,t-1}, n_k(t-1) \geq \zeta_k(T)\Big) \quad (109)$$

$$= \mathbb{P}\Big(\tilde{\mu}_{k,t-1} + \alpha_{k,t-1} \geq \tilde{\mu}_{k^\star,t-1} - \alpha_{k,t-1}, n_k(t-1) \geq \zeta_k(T)\Big) \quad (110)$$

$$= \mathbb{P}\Big(\hat{\mu}_{k,t-1} - \frac{S_k^{t-1}}{n_k(t-1)} + \alpha_{k,t-1} \geq \hat{\mu}_{k^\star,t-1} - \frac{S_{k^\star}^{t-1}}{n_k(t-1)} - \alpha_{k,t-1}, n_k(t-1) \geq \zeta_k(T)\Big) \quad (111)$$

$$\leq \sum_{l=\zeta_k(T)}^{t-1} \mathbb{P}\Big(\hat{\mu}_{k,t-1} + \frac{S_{k^\star}^{t-1} - S_k^{t-1}}{n_k(t-1)} + 2\alpha_{k,t-1} \geq \hat{\mu}_{k^\star,t-1} | n_k(t-1) = l\Big) \quad (112)$$

$$\leq \sum_{l=\zeta_k(T)}^{t-1} \mathbb{P}\Big(\hat{\mu}_{k,t-1} + \frac{M}{n_k(t-1)} + 2\alpha_{k,t-1} \geq \hat{\mu}_{k^\star,t-1} | n_k(t-1) = l\Big) \quad (113)$$

$$\overset{(a)}{\leq} \sum_{l=\zeta_k(T)}^{t-1} \mathbb{P}\Big(\hat{\mu}_{k,t-1} + \frac{\Delta_k^{\boldsymbol{\pi}}}{6} + 2\alpha_{k,t-1} \geq \hat{\mu}_{k^\star,t-1} | n_k(t-1) = l\Big) \quad (114)$$

$$\overset{(b)}{\leq} \sum_{l=\zeta_k(T)}^{t-1} \mathbb{P}\Big(\hat{\mu}_{k,t-1} + \frac{\Delta_k^{\boldsymbol{\pi}}}{6} + \frac{\Delta_k^{\boldsymbol{\pi}}}{6} \geq \hat{\mu}_{k^\star,t-1} + \alpha_{k,t-1} | n_k(t-1) = l\Big) \quad (115)$$

$$\overset{(c)}{\leq} \sum_{l=\zeta_k(T)}^{t-1} \mathbb{P}\Big(\hat{\mu}_{k,t-1} - \mu_k + \mu_{k^\star} - \hat{\mu}_{k^\star,t-1} + \frac{M}{n_k(t-1)} - \Delta_k^{\boldsymbol{\pi}} \geq \frac{-2\Delta_k^{\boldsymbol{\pi}}}{6} + \alpha_{k,t-1} | n_k(t-1) = l\Big)$$
$$\qquad (116)$$

$$\leq \sum_{l=\zeta_k(T)}^{t-1} \mathbb{P}\Big(\hat{\mu}_{k,t-1} - \mu_k + \mu_{k^\star} - \hat{\mu}_{k^\star,t-1} \geq \Delta_k^{\boldsymbol{\pi}} - \frac{3\Delta_k^{\boldsymbol{\pi}}}{6} + \alpha_{k,t-1} | n_k(t-1) = l\Big) \quad (117)$$

$$\leq \sum_{l=\zeta_k(T)}^{t-1} \Big[\mathbb{P}\Big(\hat{\mu}_{k,t-1} - \mu_k \geq \frac{\Delta_k^{\boldsymbol{\pi}}}{2} | n_k(t-1) = l\Big) + \mathbb{P}\Big(\mu_{k^\star} - \hat{\mu}_{k^\star,t-1} \geq \alpha_{k,t-1} | n_k(t-1) = l\Big)\Big] \quad (118)$$

$$\overset{(d)}{\leq} \sum_{l=\zeta_k(T)}^{t-1} \frac{1}{T^4} + \frac{1}{T^{20}} \quad (119)$$

$$\leq \sum_{l=\zeta_k(T)}^{t-1} \frac{2}{T^4} \quad (120)$$

Where $(a)$, $(b)$, and $(d)$ respectively depend on $l \geq \zeta_k(T) \geq \frac{3M}{\Delta_k}$, $l \geq \zeta_k(T) \geq \frac{162 \log T}{\Delta_k^2}$, and the Hoeffding's inequality. $(c)$ is because that $-\mu_{k^\star} + \mu_k \leq -\Delta_k^{\boldsymbol{\pi}} + \frac{M}{n_k(t-1)}$. Hence:

$$\mathbb{E}\left[n_k(T)\right] \leq \zeta_k(T) + \frac{2}{T^2} \leq \max\left\{\frac{3M}{\Delta_k^{\boldsymbol{\pi}}}, \frac{162 \log(T)}{(\Delta^{\boldsymbol{\pi}})_k^2}\right\} + \frac{2}{T^2} \quad (121)$$

$\square$

## E.2   Proof of Theorem 5.1

**Theorem 5.1.** *For any arbitrary strategy profile $\boldsymbol{\pi}$ with $M$-bounded savings, the regret of Algorithm 1 is bounded by:*

$$\mathcal{R}_T^{\boldsymbol{\pi}} = \mathbb{E}\left[\sum_{t=1}^T \left(\mu_{\sigma_{\boldsymbol{\mu}^{\boldsymbol{\pi}}}(2)}^{\boldsymbol{\pi}} - x_{k_t,t}\right) + \sum_{k=1}^K \Psi_k(x_k, x_{-k})\right]$$

$$\leq \mathcal{O}\left(\sum_{k:\sigma_{\boldsymbol{\mu}^{\boldsymbol{\pi}}}^{-1}(k)\geq 2} \max\left\{M, \frac{\log(T)}{\Delta_k^{\boldsymbol{\pi}}}\right\} + \sum_{k:\sigma_{\boldsymbol{\mu}^{\boldsymbol{\pi}}}^{-1}(k)\geq 3} \max\left\{M, \frac{\log(T)}{\underline{\Delta}_k^{\boldsymbol{\pi}}}\right\}\right) \quad (15)$$

*Proof.* Fix an arbitrary strategy profile $\boldsymbol{\pi}$. For each arm $k$, we define $S_k^T = \sum_{s=1}^T (r_{k_s,s} - x_{k_s,s}) \cdot \mathbb{1}_{[k_s=k]}$, representing the cumulative savings of arm $k$ up to round $T$. We assume that for all arms $k \in [K]$ and for any strategy $\boldsymbol{\pi}$, the cumulative savings satisfy $S_k^T \leq M$. We define the effective

mean under strategy $\boldsymbol{\pi}$ as $\mu_k^{\boldsymbol{\pi}}$, and let $\boldsymbol{\mu}^{\boldsymbol{\pi}} = (\mu_k^{\boldsymbol{\pi}})_{k \in [K]}$. Therefore, we define $\Delta_k^{\boldsymbol{\pi}}$ as the difference between the highest effective mean under strategy $\boldsymbol{\pi}$ and the effective mean of arm $k$, given by : $\Delta_k^{\boldsymbol{\pi}} = \mu_{\sigma_{\boldsymbol{\mu}^{\boldsymbol{\pi}}}(1)}^{\boldsymbol{\pi}} - \mu_k^{\boldsymbol{\pi}}$. Similarly, $\underline{\Delta}_k^{\boldsymbol{\pi}}$ represents the difference between the second-highest effective mean and the effective mean of arm $k$ under strategy $\boldsymbol{\pi}$, defined as: $\underline{\Delta}_k^{\boldsymbol{\pi}} = \mu_{\sigma_{\boldsymbol{\mu}^{\boldsymbol{\pi}}}(2)}^{\boldsymbol{\pi}} - \mu_k^{\boldsymbol{\pi}}$. The regret of Algorithm 1 with respect to the second highest effective mean can be upper bounded as in 75:

$$
\mathcal{R}_T^{\boldsymbol{\pi}} \leq \sum_{k:\sigma_{\boldsymbol{\mu}^{\boldsymbol{\pi}}}^{-1}(k) \geq 3} \underline{\Delta}_k^{\boldsymbol{\pi}} \mathbb{E}[n_k(T)] + \sum_{k:\sigma_{\boldsymbol{\mu}^{\boldsymbol{\pi}}}^{-1}(k) \geq 2} 2\Delta_k^{\boldsymbol{\pi}} \mathbb{E}\left[n_k(T)\right]
$$
$$
+ \mathbb{E}\left[(T-\tau)(\mu_{\sigma_{\boldsymbol{\mu}^{\boldsymbol{\pi}}}(2)}^{\boldsymbol{\pi}} - \tilde{\mu}_{\sigma_{\tilde{\boldsymbol{\mu}}^{\boldsymbol{\pi}}}(2)}^{\boldsymbol{\pi}})\right] + \mathbb{E}\left[\sum_{t=1}^{\tau}\sqrt{\frac{2\log(T)}{t}}\right] + K \tag{122}
$$

Hence it is crucial to upper bound $\mathbb{E}[n_k(T)]$.

By applying Lemma E.1 multiple times, we bound each component of the regret.
**First step:**

$$
\sum_{k:\sigma_{\boldsymbol{\mu}^{\boldsymbol{\pi}}}^{-1}(k) \geq 3} \underline{\Delta}_k^{\boldsymbol{\pi}} \mathbb{E}[n_k(T)] + \sum_{k:\sigma_{\boldsymbol{\mu}^{\boldsymbol{\pi}}}^{-1}(k) \geq 2} 2\Delta_k^{\boldsymbol{\pi}} \mathbb{E}\left[n_k(T)\right] \tag{123}
$$
$$
\leq \sum_{k:\sigma_{\boldsymbol{\mu}^{\boldsymbol{\pi}}}^{-1}(k) \geq 3} \underline{\Delta}_k^{\boldsymbol{\pi}} \max\left\{\frac{3M}{\underline{\Delta}_k^{\boldsymbol{\pi}}}, \frac{162\log(T)}{(\underline{\Delta}_k^{\boldsymbol{\pi}})^2}\right\} + \sum_{k:\sigma_{\boldsymbol{\mu}^{\boldsymbol{\pi}}}^{-1}(k) \geq 2} 2\Delta_k^{\boldsymbol{\pi}} \max\left\{\frac{3M}{\Delta_k^{\boldsymbol{\pi}}}, \frac{162\log(T)}{(\Delta_k^{\boldsymbol{\pi}})^2}\right\} + \frac{6K}{T^2} \tag{124}
$$
$$
\leq \sum_{k:\sigma_{\boldsymbol{\mu}^{\boldsymbol{\pi}}}^{-1}(k) \geq 3} \max\left\{3M, \frac{162\log(T)}{\underline{\Delta}_k^{\boldsymbol{\pi}}}\right\} + \sum_{k:\sigma_{\boldsymbol{\mu}^{\boldsymbol{\pi}}}^{-1}(k) \geq 2} 2\max\left\{3M, \frac{162\log(T)}{\Delta_k^{\boldsymbol{\pi}}}\right\} + o(1) \tag{125}
$$

**Second step:**

$$
\mathbb{E}\left[\sum_{t=1}^{\tau}\sqrt{\frac{2\log(T)}{t}}\right] \leq \sqrt{8\log(T)}\mathbb{E}[\sqrt{\tau}] \tag{126}
$$
$$
\leq \sqrt{8\log(T)}\sqrt{\mathbb{E}\left[\tau\right]} \quad \text{(concavity of square root function)} \tag{127}
$$
$$
\leq \sqrt{8\log(T)}\sqrt{2\sum_{k:\sigma_{\boldsymbol{\mu}^{\boldsymbol{\pi}}}^{-1}(k) \geq 2} \mathbb{E}\left[n_k(T)\right]} \tag{128}
$$
$$
\leq \sqrt{8\log(T)}\sqrt{2\sum_{k:\sigma_{\boldsymbol{\mu}^{\boldsymbol{\pi}}}^{-1}(k) \geq 2} \left[\max\left\{\frac{3M}{\Delta_k^{\boldsymbol{\pi}}}, \frac{162\log(T)}{(\Delta_k^{\boldsymbol{\pi}})^2}\right\} + \frac{2}{T^2}\right]} \tag{129}
$$
$$
\leq \sqrt{8\log(T)}\sqrt{2\sum_{k:\sigma_{\boldsymbol{\mu}^{\boldsymbol{\pi}}}^{-1}(k) \geq 2} \max\left\{\frac{3M}{\Delta_k^{\boldsymbol{\pi}}}, \frac{162\log(T)}{(\Delta_k^{\boldsymbol{\pi}})^2}\right\} + \frac{4K}{T^2}} \tag{130}
$$
$$
\leq \sqrt{16\log(T)}\sum_{k:\sigma_{\boldsymbol{\mu}^{\boldsymbol{\pi}}}^{-1}(k) \geq 2} \sqrt{\max\left\{\frac{3M}{\Delta_k^{\boldsymbol{\pi}}}, \frac{162\log(T)}{(\Delta_k^{\boldsymbol{\pi}})^2}\right\}} + o(1) \tag{131}
$$
$$
\overset{\text{Fact A.2}}{\leq} \sum_{k:\sigma_{\boldsymbol{\mu}^{\boldsymbol{\pi}}}^{-1}(k) \geq 2} \max\left\{3M, \frac{162\log(T)}{\Delta_k^{\boldsymbol{\pi}}}\right\} + o(1) \tag{132}
$$
$$
\tag{133}
$$

**Third step:**

$$
\mathbb{E}\left[(T-\tau)(\mu_{\sigma_{\boldsymbol{\mu}^{\boldsymbol{\pi}}}(2)}^{\boldsymbol{\pi}} - \tilde{\mu}_{\sigma_{\tilde{\boldsymbol{\mu}}^{\boldsymbol{\pi}}}(2)}^{\boldsymbol{\pi}})\right] = o(1) \quad \text{(similar arguments 86–92)} \tag{134}
$$

Combining the three steps gives:

$$\mathcal{R}_T^{\boldsymbol{\pi}} \leq \mathcal{O}\left( \sum_{k:\sigma_{\boldsymbol{\mu}^{\boldsymbol{\pi}}}^{-1}(k)\geq 2} \max\left\{ M, \frac{\log(T)}{\Delta_k^{\boldsymbol{\pi}}} \right\} + \sum_{k:\sigma_{\boldsymbol{\mu}^{\boldsymbol{\pi}}}^{-1}(k)\geq 3} \max\left\{ M, \frac{\log(T)}{\underline{\Delta}_k^{\boldsymbol{\pi}}} \right\} \right) \tag{135}$$

$\square$

## F   Empirical Analysis

We conducted an experiment with six arms, where the rewards followed the order ($\mu_1 > \mu_2 > \mu_3 > \mu_4 > \mu_5 > \mu_6$). The experiment spanned a horizon of $10^4$ time steps and was averaged over 100 epochs. The game's dynamics are inherently complex, as arm strategies are typically intractable in practical scenarios. To gather empirical evidence, we fixed certain strategies and monitored the player's regret while using Algorithm 1. Our study examined three specific scenarios:

1. **Untruthful Arbitrary Reporting:** At each round, the selected arm randomly reports 100%, 60%, 40%, 10%, or 0% of its observed reward.

2. **Truthful Reporting:** This corresponds to the Dominant SPE.

3. **"Optimal" Reporting:** Here we use the term "optimal" somewhat loosely. In this scenario, only the two best arms report truthfully, while the remaining suboptimal arms report 0.

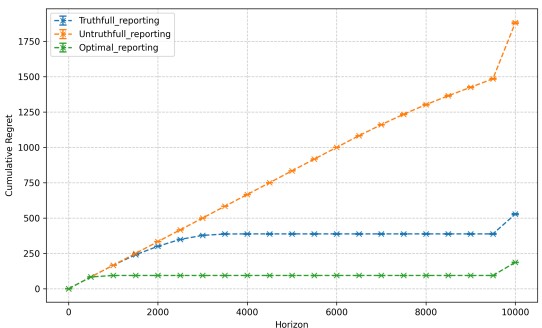

Figure 1: Cumulative regret under different strategies: **1.** Under untruthful arbitrary reporting, where arms arbitrarily choose to keep a portion of their reward. **2.** Under truthful reporting, where arms adhere to the dominant truthful SPE. **3.** Under "optimal" reporting, where only the two best arms report truthfully and the remaining suboptimal arms withhold the entirety of their observed reward.

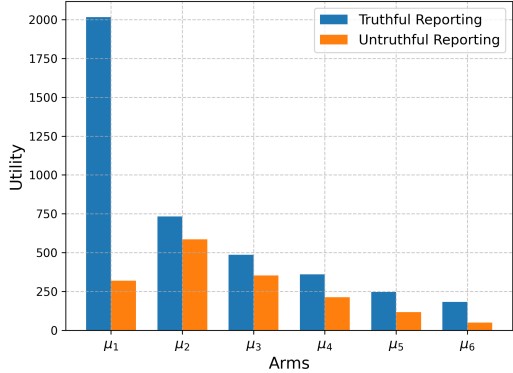

Figure 2: Comparison of arms' utilities between truthful reporting and arbitrary untruthful strategies

In Figure 1, we present the cumulative regret of the player for each strategy, where the final point of each curve represents the bonus paid to the arms at the end of the game. The results are averaged over

multiple epochs. As expected, the worst regret is observed in the first scenario, where arms withhold an arbitrary portion of the observed reward. For the two remaining scenarios, both exhibit logarithmic cumulative regret, with the "optimal" reporting scenario demonstrating a better factor. This is because the algorithm eliminates suboptimal arms more quickly and transitions faster to the second phase where it achieves better performance. While "optimal" reporting is challenging to achieve, **S-SE** effectively incentivizes arms to report truthfully, creating a dominant SPE and guaranteeing the player logarithmic regret.

Within the same experiment, we calculated the total gains for each arm under two scenarios: truthful reporting (i.e., the dominant Subgame Perfect Equilibrium, SPE) and arbitrary untruthful reporting, where arms misreport a portion of the observed rewards. As shown in Figure 2, we observed that under truthful reporting, the arms' utilities were higher compared to those under untruthful reporting, consistent with the theoretical predictions.

