# OpenReview forum: "Strategic Multi-Armed Bandit Problems Under Debt-Free Reporting"
_NeurIPS.cc/2024/Conference — NeurIPS 2024 poster_

### Official Review · Reviewer_VQPm · 2024-07-08

**Soundness:** 3
**Presentation:** 3
**Contribution:** 2
**Rating:** 6
**Confidence:** 4

**Summary:**

This paper considers a strategic variant of the multi-armed bandit problem with payments. It thereby builds upon a problem studied by Braverman et al. The paper formally introduces the problem formulation and proposes an algorithm, S-SE, that combines successive elimination with a meticulously chosen payment rule. It is shown that truthfulness is a dominant SPE under which the proposed algorithm suffers logarithmic regret. The paper also analyzes the utility of the arms under S-SE and the dominant SPE. Finally, the performance of the algorithm under arbitrary arm strategies is analyzed.

**Strengths:**

- The studied problem, which is at the intersection of bandit learning and mechanism design, is very interesting.
- The paper is well-written and easy to follow.
- It is interesting to see that the regret achieved by Braverman et al. can be improved using an extension of successive elimination + payment rule (under debt-free reporting).
- The paper proves that truthfulness is a *dominant* SPE.
- The provided regret bound appears to be near-optimal and the dependence on the gaps $\Delta_{1k}$ and $\Delta_{2k}$ is quite interesting.
- Overall, even though many aspects of this problem have already been studied by Braverman et al., I believe that this papers makes some insightful and novel contributions.

**Weaknesses:**

- The paragraph **Tightness of Regret** is a very hand-wavy. In strategic settings like the one you study here, one easily mistakes an intuition about "rational agent behavior" for a rigorous argument. In particular, you write
 "The optimal arm only needs to report a slightly higher value than the second-best arm to be chosen by the player as the superior option [*you here assume a class of mechanisms that do in fact choose the arm with highest reported value in some specific way*]. Consequently, **all efficient low-regret mechanisms** must ensure truthfulness, at least for the two best arms."
Intuitively, yes, I agree. However, this is a claim that, I think, is very difficult to prove rigorously. I recommend to adapt the language used in this paragraph and emphasize that you're providing intuition only.
- Similar hand-wavy language and statements are used a few times in the text (not as bad as the above one). Please make sure to be clear whether you provide intuition (which is great) or you actually proved the statement (even better).
- A minor thing that is also about the use of slightly imprecise language and might come across as nitpicking. Lemma 14 in Braverman et al. actually shows that you cannot achieve $(\alpha \mu_1 + (1-\alpha) \mu_2)T$ utility for *constant* $\alpha > 0$. Hence, you can achieve more than $\mu_2 T$ revenue, e.g., let $\alpha = 1/\sqrt{T}$, despite you saying in line 126 that "no mechanism can guarantee more than $\mu_2 T$ revenue". This is not a serious issue, since I think we can all agree that defining regret w.r.t. $\mu_2 T$ is still the right choice in view of Lemma 14. I still wanted to make you aware of this.

**Questions:**

- Line 269: You say that the best arm is incentivized with a bonus of $O(\Delta_{12} T)$. Is this a typo? If the bonus payment would be of order $T$, then your regret bound would be of order $T$. It also doesn't seem necessary to make such a large bonus payment, since you'd only need to compensate the best arm for its truthful reporting in the first $\tau$ rounds.


**Minor things:**
- Consider splitting the text in Section 2 into several paragraphs to improve readability. Also, clearly highlighting the observational model and assumptions using \paragraph or some kind of \emph would be helpful, in my opinion.
- Typos: "tof", "understrategy"

**Limitations:**

Limitations are adequately addressed in my opinion.

---

> ### Author Rebuttal · Authors · 2024-08-06
>
> Dear Reviewer VQPm,
>
> Thank you very much for your detailed and insightful review. We greatly appreciate the time and effort you dedicated to it.
>
> We are computing regret with reference to $\mu_{2}$, which involves incentivizing the best arm with $O(T \Delta_{12})$. This approach is not harmful, as it offsets the difference between $\mu_{1}$ and $\mu_{2}$ while preserving $\mu_{2}$ for the player. Essentially, we allocate all values greater than $\mu_{2}$ to the best arm throughout the game. Additionally, the incentive should address the second phase. If not properly incentivized during this phase, the best arm might withhold a portion of the reward (especially since the second-best mean is not communicated), leading to reported values that are less than $\mu_{2}$ on average. This could result in linear regret, given that the second phase lasts much longer than the first. Thus, incentivizing the best arm with $O(T \Delta_{12})$ appears necessary (as also done by Braverman et al.), which is acceptable given the definition of regret. However, improvements have been achieved through the new bonus structure assigned to suboptimal arms, which has been optimized due to the adaptiveness of the new algorithm.
>
> We appreciate your feedback on the weaknesses in our representations, language, and typos. We will address these issues and make the necessary corrections in the final version.
>
> We hope that we have addressed your concerns and answered your questions clearly, contributing positively to your evaluation.

---

> > ### Comment · Reviewer_VQPm · 2024-08-07
> >
> > Thanks for clarifying. When I wrote the questions I forgot that your benchmark is the second best arm.
> >
> > I decide to keep my original score with the primary reason being that, even though the paper makes some novel contributions, these are to some degree incremental. I'm still in favor of acceptance.

---

> > > ### Author Response · Authors · 2024-08-08
> > >
> > > Thank you very much for your valuable feedback and the insightful discussion.

---

### Official Review · Reviewer_3dvn · 2024-07-11

**Soundness:** 3
**Presentation:** 3
**Contribution:** 2
**Rating:** 8
**Confidence:** 4

**Summary:**

Multi-armed bandit problems capture explore-exploit scenarios under different reward structures including stochastic and adversarial. In this paper, the authors consider bandit problems where the arms report rewards strategically. To tackle such a problem, the authors devise a successive elimination scheme, where arms are eliminated till the best arm is identified (with high probability). To encourage truthful reporting, the authors devise a bonus scheme, so that the best arm is pulled unless it dips below the second-best arm, which encourages truthful reporting. In scenarios where arms fail to report truthfully, the authors develop regret bounds detailing the performance of algorithms.

**Strengths:**

1. **Problem Formulation is Interesting** - The authors investigate debt-free reporting under the scenario of a bonus structure. The problem structure/formulation is interesting as it covers incentives, in the form of bandits reporting arms strategically, and the explore-exploit dilemma found across bandit works.
2. **Dominant Strategy is Detailed and Analyzed** - In Section 4, the authors characterize the dominant strategy for arms, under a certain type of bonus payoff, and show that this results in truthful reporting from arms. Such a characterization helps us better understand how no-regret algorithms can be derived and provides an important piece of information on these types of bandits.
3. **Regret Bounds in both Strategic and Non-Strategic Scenarios** - The authors derive regret bounds in both strategic and non-strategic scenarios. For non-strategic scenarios, the truthfulness of the arms makes it easier to determine optimal arm pulls, while for non-truthful scenarios, they use the upper bound on savings, M, to characterize regret. It would be nice to understand how the regret bound proved in Theorem 5.1 corresponds to some type of function of T.
4. **Bonus Structure as Payment Improves Regret Bounds** - Table 1 lists a comparison between the method proposed in their work (S-SE) and previous work (S-ETC). The two models differ in their bonuses, and they show that by modifying this, they go from a T^2/3 bound to a T^1/2 bound.

**Weaknesses:**

1. **Lack of Empirical Verification** - On page 8, the authors discuss the tightness of the regret bounds, and argue that while the actual regret might change, the order of the regret is roughly the same across scenarios. However, it would be interesting to understand how tight such regret bounds are across different scenarios. Under what types of scenarios are the regret bounds tight and under which are regret bounds loose? An empirical analysis, with even simulated data, would provide a nice complement to the theoretical analysis proposed in this paper.

**Questions:**

1. Could you further detail the types of scenarios or real-world regimes where such bandits exist?

**Limitations:**

The future work section discusses additional questions that could be answered, though authors should more explicitly label this as a limitations section

---

> ### Author Rebuttal · Authors · 2024-08-06
>
> Dear Reviewer 3dvn,
>
> Thank you for your insightful feedback and for the time you dedicated to reviewing our paper.
>
> In the current version of our paper, we did not include experimental results, which is indeed quite common in game theory, due to the fact that the dynamics of the game are intrinsically dependent on arm strategies that are generally intractable in real-world scenarios. Consequently, we provided a theoretical analysis of the dominant Subgame Perfect Equilibrium (SPE) and discussed the tightness of its regret. However, inspired by your review, we have conducted a simulated experiment where we fix the arm strategies from the beginning and track the cumulative regret. Specifically, we consider six strategic arms with $\mu_{1} > \mu_{2} > \mu_{3} > \mu_{4} > \mu_{5} > \mu_{6}$. The experiment is run over a horizon of 10,000 and averaged over 100 epochs. We examine three scenarios:
>
> 1. **Untruthful Arbitrary Reporting**: At each round, the selected arm randomly reports 100%, 60%, 40%, 10%, or 0% of its observed reward.
> 2. **Truthful Reporting**: This corresponds to the Dominant SPE.
> 3. **"Optimal" Reporting**: Here we use the term "optimal" somewhat loosely. In this scenario, only the two best arms report truthfully, while the remaining suboptimal arms report 0.
>
> The results are detailed in the attached PDF. As expected, the worst regret corresponds to the first scenario. However, for the two remaining scenarios, both exhibit logarithmic cumulative regret, with the "optimal" reporting scenario demonstrating a better factor since the algorithm eliminates suboptimal arms more quickly and transitions faster to the second phase where it achieves better performance. While "optimal" reporting is challenging to achieve, **S-SE** effectively incentivizes arms to report truthfully, creating a dominant SPE and guaranteeing the player logarithmic regret.
>
> Our model is inspired by several real-world applications and can be seen as a multi-agent extension of the principal-agent problem in contract theory. It deals with dynamic agency issues where a principal must select one of K agents to carry out tasks on their behalf, with the cost of these tasks remaining unknown to the principal (e.g., choosing among K contractors for a job). A crucial aspect is that the principal does not know in advance the exact cost or benefit of the actions performed. The concept of debt-free reporting of strategic arms is inspired by e-commerce transactions, where a platform may choose to cancel a sale but cannot create one. This generalizes to binary bandits, where it is easy to hide a success and report a fake failure but difficult to create a success. Additionally, it is motivated by repeated trades with budget constraints, where an arm cannot report more than it possesses.
>
> We also agree with your suggestion regarding the relabeling of the limitations section and will make the necessary corrections.
>
> We hope that we have addressed your concerns and answered your questions satisfactorily, contributing positively to your evaluation.

---

> > ### Comment · Reviewer_3dvn · 2024-08-07
> > **Thank you for your new experiments**
> >
> > Thank you for your clarifications and the new experiments. I am satisfied with these experiments and am happy to raise my score to an 8.

---

> > > ### Author Response · Authors · 2024-08-08
> > >
> > > Thank you very much for your valuable feedback and the insightful discussion.

---

### Official Review · Reviewer_JBtJ · 2024-07-13

**Soundness:** 3
**Presentation:** 3
**Contribution:** 3
**Rating:** 6
**Confidence:** 3

**Summary:**

The paper addresses a strategic bandit setting. Specifically, the player (algorithm) can select between K arms as in standard bandits but each arm can choose the reward it reports instead. Because of the debt-free assumptions, the reported value cannot exceed the realized reward. The papers gives an algorithm that rewards the arms based on the values they report.  The regret is defined with respect to the second highest mean + the total payment given to the arms. The given algorithm can achieve logarithmic regret which is an improvement over the regret of [8].

**Strengths:**

-The paper essentially resolves an open problem of [8] where the arms are only debt-free.

-The paper also includes an additional bound which nicely characterizes the regret under deviations.

**Weaknesses:**

-The motivation of the paper does not feel strong. Specifically, the paper answers a small question from reference [8].

-While [8] rewards using rounds, here the reward is through payment which gives higher flexibility. Can the algorithm be generalized to allow payments through rounds instead (assuming a fixed horizon T)?


-If the player knows the minimum gap $\Delta$ can an explore then commit based algorithm similar to [8] be used to achieve log T regret?

Below are Minor representation issues:


-typo: t → $t$ in line 17






-Bullet 2 in the contribution: the regret having $\Delta_{2k}$ suggests that we should instead have $\mu_2 > \mu_3 $, i.e. strict inequality.



-There is a latex problem  for algorithm 1 block, it refers to the model on page 2 instead.

**Questions:**

Please adress the above points under weaknesses.

**Limitations:**

Limitations are adequately addressed.

---

> ### Author Rebuttal · Authors · 2024-08-02
>
> Dear Reviewer JBtJ,
>
> We appreciate your review and the time you've dedicated to it.
>
> In this paper, we concentrate on strategic arms under debt-free reporting, driven by various real-world applications such as interactions on e-commerce platforms and repeated trades with budget constraints [9, 10], where reported values cannot surpass observed ones. Under the unconstrained payment setting, where arms can report any value, [8] presented an optimal mechanism with constant regret. However, when this assumption is relaxed to restricted payments (i.e., debt-free reporting), [8] proposes a mechanism that suffers from a regret of $T^{2/3}$. Hence, moving from unrestricted payment to debt-free reporting increases the regret from a constant value to $T^{2/3}$. This observation motivated us to focus on this setting.
>
> While [8] employs additional rounds at the end of the game as incentives, we propose using payments at the end. Both types of bonuses are allocated at the end of the game, and bonuses are subtracted from player revenue and considered in the regret calculation, regardless of their nature. By using payments as incentives, defining truthful reporting as a dominant subgame perfect equilibrium (SPE) is straightforward. However, with additional rounds, we refer to a pseudo-truthful strategy since truthfulness applies only to rounds excluding the additional bonus rounds, specifically during the initial phase when the algorithm is learning the statistics of each arm. During the additional bonus rounds, which occur at the end of the game, the player ignores the values reported by the arms, as clearly stated in [8]. Given that actual payment is a common practice in typical mechanism designs for procurement auctions, we chose actual payments to maintain a more direct intuition and facilitate the presentation. In fact, a modified version of our algorithm that includes playing each arm $\frac{\Psi_k}{\mu_k}$ at the end effectively regains the payment through rounds. As already mentioned in Section 6, the improvement in regret does not depend on the type of incentives, whether they are additional bonus rounds or bonus payments. This is because bonuses are subtracted from player revenue and considered in the regret calculation, irrespective of their nature. Therefore, replacing the additional rounds in [8] with payments will not lead to an improvement in regret. The improvement is mainly due to the adaptivity of the algorithm, which allows for more tailored and less harmful bonuses. However, this adaptivity also necessitates a more technical solution concept due to the extensive nature of the game, as opposed to the simultaneous one used in [8].
>
> If the player knows the minimum gap $\Delta$, an explore-then-commit-based algorithm similar to [8] can achieve $\log T$ regret. In fact, it suffices to explore each arm for $\log(T)/\Delta^2$ rounds instead of $T^{2/3}$. However, the purpose of bandit algorithms is to overcome the lack of this knowledge and effectively address cases where $\Delta$ is unknown.
>
> We thank you for pointing out the representation issues, and we assure you that we will adjust the final version of our work accordingly.
>
> We hope this addresses your concerns and contributes positively to your evaluation.

---

### Official Review · Reviewer_byTp · 2024-07-14

**Soundness:** 4
**Presentation:** 3
**Contribution:** 2
**Rating:** 6
**Confidence:** 4

**Summary:**

This paper studies a multi-armed bandit setting with stochastic arms but where the arms are strategic agents -- when an arm is pulled it can choose what fraction of the reward to keep for itself and what fraction of the reward to pass on to the principal (the learning algorithm picking the arms). The principal can only see the reward they receive (not the amount received by the agent), and all parties want to maximize their total reward.

This setting was originally studied by Braverman et al., who showed that if the principal runs a standard no-regret learning algorithm then there are subgame perfect “colluding” equilibria where the principal receives nothing -- on the other hand, the principal can always asymptotically achieve the second highest average payoff by essentially running a second-price auction at the beginning of the game and only pulling the winner.

This strategy proposed by Braverman et al. has two issues. One is that (naively implemented) it may require agents (the arms) to report a higher value in some early rounds than the value they actually achieve. Even if you fix this issue (by converting this idea into an explore-then-commit style algorithm), the regret of the principal -- the gap between their expected utility and actually receiving the second-highest average per round -- will be a suboptimal O(T^{2/3}).

This paper shows how to get O(sqrt(TlogT)) instance-independent and O((log T)/Delta) instance-dependent regret bounds for this problem, on par with regret bounds achievable for standard bandit problems. The main idea is to adapt the Successive Elimination bandit algorithm (and its analysis) to this strategic setting. At a high level, the algorithm runs SE until there is only arm left, and then requires this arm to contribute (on average) at least the second-highest average reward per round. Finally, all participating agents are given a bonus designed to make bidding their true value incentive compatible.

**Strengths:**

Understanding how to adapt standard learning algorithms to strategic settings is an important question. This paper provides an improved solution to a natural strategic learning environment and presents (what I would guess to be) tight regret bounds for learning in this environment. Although the algorithm is an adaptation of a standard no-regret algorithm (successive elimination), analyzing its performance in this strategic environment is non-trivial (as it involves pinning down the worst possible equilibrium behavior of the individual arms). The paper was well-written and easy to read.

**Weaknesses:**

I think it is a little unsatisfactory that the optimal learning algorithm still takes the form “auction off the business to the best-arm for the second-best-arm’s price”, even if this auction is now done in a slightly smoother way (via successive arm elimination over several rounds, instead of a sublinear exploration period where arms simply reveal their price). I.e., both this algorithm and the previous algorithm have the property that after some stage of the game, they will only ever select one specific arm. I think it would be a more interesting result if the principal’s strategy was more time-stable and could e.g. handle arms entering or leaving the market (or adversarial rewards, although that would require more changes to the problem set-up).

Another aspect of the solution that bothers me a little bit is that the new scheme requires bonuses that grow linearly in T. This seems to require a lot of trust on behalf of the agents that they will in fact eventually get reimbursed. In contrast, the previous algorithm of Braverman et al. also implements some bonus scheme (through rewarding the agents in some extra rounds at the end of the game), but these bonuses are sublinear in the time horizon.

**Questions:**

Is it possible to modify the S-SE algorithm to work with sublinear bonuses?

Do the regret lower bounds from standard bandits carry over? (I.e., are the regret bounds in this paper tight?).

Feel free to reply to any other aspect of the review.

**Limitations:**

Limitations are adequately addressed.

---

> ### Author Rebuttal · Authors · 2024-08-02
>
> Dear Reviewer byTp,
>
> We express our sincere gratitude for your invaluable review and the time you have dedicated to it.
>
> This paper, motivated by various real-world applications such as e-commerce and repeated trades with balanced budgets, delves into strategic arms under debt-free reporting. Our primary objective is to address an open problem in [8], which involves improving the regret under debt-free reporting, aiming to progress from $T^{2/3}$ to $\sqrt{T \log(T)}$. Therefore, we adopt the same setting as in [8], including a fixed number of strategic arms. On one hand, this setting inherently includes the scenario of arms exiting the market, equivalent to arms deciding to report 0 for all subsequent steps. On the other hand, accommodating arms entering the market would necessitate significant changes to the setting (compared to [8]) and would require introducing concepts of adversarial rewarding or even non-stationary systems. While we agree that this is a fascinating proposal, we demonstrate that even within the same setting as [8], there are substantial improvements (such as a more suitable incentive-compatible algorithm, better/tighter regret bounds, robustness to small deviations, etc.), which form the core of this paper. We assure you that the notion of a time-variant market will be included in the future work section, as we recognize that it will motivate further significant research.
>
> In the strategic setting, under the most unfavorable circumstances, it is impossible to guarantee gains surpassing $\mu_2$. This limitation arises because the optimal arm only needs to report a marginally higher value than the second-best arm to be chosen as the superior option by the player, as formally presented in Lemma 14 of [8]. Therefore, providing a bonus of $O(T \Delta_{12})$ to the best arm to ensure truthful reporting remains optimal for the player, as it guarantees a return of $\mu_{2}$ each time the best arm is played. Generally speaking, the harmful bonus in the strategic case is the one subtracted from $\mu_{2}$, i.e., the ones paid to the suboptimal arms, which is sublinear in our case (on the order of $\log(T) / \Delta$). As a side note, even [8] rewards the best arms with $O(T \Delta_{12})$ (refer to Mechanism 2, Line 3), which is entirely acceptable since it does not contribute to the regret.
>
> In the paragraph "Tightness of Regret" (line 271 of our paper), we discussed that the upper bound on the regret is tight and that the lower bound is of the same order as the result given in (12).
>
> Additionally, we want to highlight the contributions of our paper. There is a notable technicality worth mentioning regarding the definition and proof of our solution concept. Our setting involves an extensive game, as opposed to a simultaneous one (as in Braverman et al., 2019, where learning occurs once at the beginning). In our extensive game, the equilibrium strategy of each arm at each node must consider the updated history of the entire game up to that point. This task is generally intractable (Ben-Porath, 1990; Conitzer and Sandholm, 2002). Interestingly, we demonstrate that our mechanism ensures truthfulness as the best response to adversarial strategies independent of history. Our approach entails consistent truthful reporting, regardless of the equilibrium path. Notably, this remains an equilibrium irrespective of past events, resembling sub-game perfect equilibrium. While the adaptivity of our algorithm introduces a more complicated solution concept for the analysis, it is crucial for the improvement in regret that we present. This adaptivity necessitates well-tailored and less harmful incentivizing bonuses, which are key to better regret. The paper also includes an additional bound which nicely characterizes the regret under deviations  giving an idea about the robustness of the algorithm.
>
> We appreciate your feedback and assure you that this discussion will be incorporated into the final version of our work. We hope this addresses your concerns and contributes positively to your evaluation, leading to a better score.

---

> > ### Comment · Reviewer_byTp · 2024-08-12
> >
> > Thank you for the detailed response. After reading the response and other reviews, I have decided to increase my score slightly.

---

> > > ### Author Response · Authors · 2024-08-14
> > >
> > > Thank you very much for your valuable feedback and the insightful discussion.

---

### Author Rebuttal · Authors · 2024-08-06

Here we present the complementary experimental results to our theoretical analysis, as suggested by Reviewer 3dvn.

---

### Decision · Program_Chairs · 2024-09-25

**Decision:**

Accept (poster)

**Comment:**

Reviewers are all positive about the paper. They mention that the paper studies an interesting and important problem and resolves an open problem of prior work Braverman et al.. Reviewers also have some comments on writing and it would be good to address them in the camera ready version.